# Modelling the barotropic sea level in the Mediterranean Sea using data assimilation

Marco Bajo[1], Christian Ferrarin[1], Georg Umgiesser[1,2], Andrea Bonometto[3], and Elisa Coraci[3]

[1]Institute of Marine Sciences, National Research Council, Castello 2737/F, 30122 Venice, Italy
[2]Klaipėda University, Coastal Research and Planning Institute, H.Manto 84, 92294 Klaipėda, Lituania
[3]Italian Institute for Environmental Protection and Research, S. Marco, 4665, 30122 Venice, Italy

**Correspondence:** Marco Bajo (marco.bajo@ve.ismar.cnr.it)

**Abstract.** This paper analyses the variability of the sea level barotropic components in the Mediterranean Sea and their reproduction using a hydrodynamic model with and without data assimilation. The impact of data assimilation is considered both in reanalysis and short-forecast simulations. We used a two-dimensional finite element model paired with an ensemble Kalman Filter, which assimilated hourly sea-level data from 50 stations in the Mediterranean basin. The results show a significant improvement given by data assimilation in the reanalysis of the astronomical tide, the surge and the barotropic total sea level, even in coastal areas and far from the assimilated stations (e.g., the south-eastern Mediterranean Sea). As the reanalysis simulations, the forecast simulations, which start from analysis states, improve especially on the first day (37% average error reduction) and when seiche oscillations are triggered. Since seiches are free barotropic oscillations depending only on the initial state, their reproduction improves very effectively with data assimilation. Finally, we estimate the periods and the energy of these oscillations by means of spectral analysis, both in the Adriatic Sea, where they have been extensively studied, and in the Mediterranean Sea, where the present documentation is scarce. While the periods are well reproduced by the model even without data assimilation, their energy shows a good improvement using it.

## 1 Introduction

Due to its historical and geopolitical importance, the Mediterranean Sea (Fig. 1) is extensively studied from every point of view, including the physical one. Marine circulation, the main physical, chemical and biological parameters are the subject of numerous researches at various spatial and temporal scales. As regards the sea level, the most extreme phenomena, which are caused by meteorological storms in conjunction with astronomical tide (Cavaleri et al., 2019; Ferrarin et al., 2021), happen often in the northern Adriatic Sea (Fig. 1). In the rest of the Mediterranean basin, these phenomena are less frequent and, usually, the sea level variations are studied on a longer time scale, linked to the baroclinic circulation. However, even in these parts of the Mediterranean Sea, barotropic variations of the sea level of few hours and tens/hundreds of kilometres, have a certain importance. They can be divided, according to their forcing, into astronomical tide, surges and seiches (Pugh, 1996).

In the central and northern Adriatic Sea, the shallowness of the continental shelf favours the growth of sea level perturbations. Indeed, the northern Adriatic Sea is one of the Mediterranean regions (as the Gulf of Gabes) experiencing the highest tidal oscillations (about 1 m at spring tide; Tsimplis et al., 1995). Concerning the surge, the presence of strong autumn south-easterly

winds (Scirocco), blowing along the main axis of the basin, favours storm surge events in the north; events that can trigger seiche oscillations of considerable intensity (Međugorac et al., 2016). Therefore, the floods in the northern Adriatic coasts but also in the rest of the Mediterranean coasts, consist of a superimposition of astronomical tide, surges and pre-existing seiches, which are generated by previous storm surge events. In densely populated cities with important cultural heritage, such as Venice and Dubrovnik in the Adriatic or Alexandria in the south-eastern Mediterranean basin, it is essential to provide a correct forecast of the sea level at short lead time, from nowcasting up to about five days ahead, to alert the population and the authorities of possible flooding events. In this time window, tides, surges and seiches are the main components influencing the sea level variations. Other possible variations of the sea level related to violent storms could be due to river run-offs, but this component is negligible in the Mediterranean Sea. As asserted before, these phenomena are stronger in the Adriatic Sea, but sometimes the western Mediterranean is subject to strong Mistral events (north-westerly winds) and the southern Mediterranean shows the formation of small but intense cyclones with tropical dynamics (called medicanes). These extreme weather conditions have already caused flooding events in the past, even in areas traditionally not affected (Scicchitano et al., 2021).

Regarding the seiches, they are triggered by surge events and have periods determined by the barotropic modes of a basin. While the modes of the Adriatic Sea, being very energetic, have been well studied in the past, those of the Mediterranean Sea are not well known. Although a correct reproduction of seiche oscillations is mandatory in the Adriatic in case of extreme events, also in normal condition it improves the sea level both in the Adriatic and in the Mediterranean. Furthermore, the investigation of the normal barotropic modes of a basin can be interesting also due to the fact that these modes can be triggered by tsunami waves.

The predictability of tides and surges depends on the predictability of their forcings. The astronomical tide, due to its periodic nature, can be predicted with good accuracy where in-situ sea-level observations are available. Where these observations lack, the tide can be computed by altimeter data (Birol et al., 2017) or using a hydrodynamic model (with good bathymetry data). As regards the surge, in case of severe weather conditions, most of the error on the sea level is due to this part. The surge has a non-periodic nature, depending on the surface wind and atmospheric pressure and, if the meteorological forcing is wrong, the errors can be consistent (Barbariol et al., 2022). Surges can trigger seiches, which propagate for several days as well as their errors, with different periods and decay times, depending on the barotropic modes which they follow. To reduce the errors of these sea-level components, data assimilation (DA) procedures can be used. DA aims to reduce the error of the state of a dynamic model at a fixed time by exploiting the available observations of quantities correlated to the model's variables (Kalnay, 2002; Evensen, 2009a; Carrassi et al., 2018). DA can be used both to improve the forecast, providing an accurate initial state, which is called *analysis state* or to produce several analysis states to simulate past periods with reduced errors (*reanalysis simulation*). Usually, the reanalysis simulations are much more accurate than analogous simulations made without the DA (here referred to as *hindcast simulations*). This is due to not only to the assimilation of all the available well-processed observations, but also, if possible, to the use of more accurate forcings and boundary conditions.

The main purpose of this work is to analyse the impact of DA in the reproduction of tides, surges, seiches and the total sea level made by these components, both in reanalysis and in forecast simulations. As regards the astronomical tide, the reanalysis

simulation can be used to produce more accurate maps of the amphidromic systems of its components. Moreover, harmonic analyses can be executed at each point of the model's grid to determine the amplitudes and phases of the main components in order to obtain forecasts in arbitrary locations. The reanalysis of the surge and the total sea level, is useful mainly as a coastal product, to produce accurate past climatologies with a good reproduction of extreme events. For example, in the Mediterranean Sea, where the coasts have a large extension compared to the basin's area and the weather conditions are strongly influenced by the orography, hindcast model simulations, without DA, often suffer from underestimation errors.

The DA can be used not only for the reanalysis but also for the forecast, by improving the initial state of the system. In this work we use the DA one day before each daily forecast, to create a final state of analysis from which to start the forecast simulation. The DA improvement is due to the fact that the initial analysis state has a lower error than the one without DA (*background state*), even if the error coming from the forcing and boundary condition cannot be corrected. In summary, we run the simulations using a finite element hydrodynamic model and assimilating, in some of them, data from 50 sea-level coastal stations with an ensemble Kalman filter. The period considered is two-month long from the beginning of November to the end of December 2019. In this period one of the most extreme storm surge event was recorded in Venice and very energetic seiche oscillations happened some weeks later.

In the following sections, we report the methodology, with a description of the hydrodynamic model (Section 2.1), the observation collection and processing (Section 2.2) and the DA method and setup (Section 2.3). The section ends with a description of all the simulations that we performed (Section 2.4). Then, we expose the results of the DA calibration (Section 3.1), the hindcast/reanalysis simulations (Section 3.2) and the forecast simulations (Section 3.3). The second part of Section 3.3 is dedicated to the description and reproduction in the forecast mode of the November and December 2019 extreme events mentioned before. Finally, the discussion (Section 4) and conclusions (Section 5) follow.

## 2  Methods

### 2.1  The hydrodynamic model

The hydrodynamic model we use is called SHYFEM (System of HydrodYnamic Finite Element Module - v7_5_74) and was created at the CNR in Venice (Umgiesser and Bergamasco, 1993), where it is largely developed. Its code is available with open-source license and freely downloadable from the Web (https://github.com/SHYFEM-model/shyfem). SHYFEM is composed of a hydrodynamic core that solves the shallow water equations with the finite element technique and with a semi-implicit time-stepping algorithm, which allows a remarkable speed of execution. Various terms in the equations can be turned on or off, such as the momentum advection terms, Coriolis terms, baroclinic terms and tidal potential. The model can be used in two- or three-dimensional modes and allows various formulations of bottom stress and wind stress. Finally, the model can be coupled to various modules or other models (e.g., waves, Lagrangian, ecological).

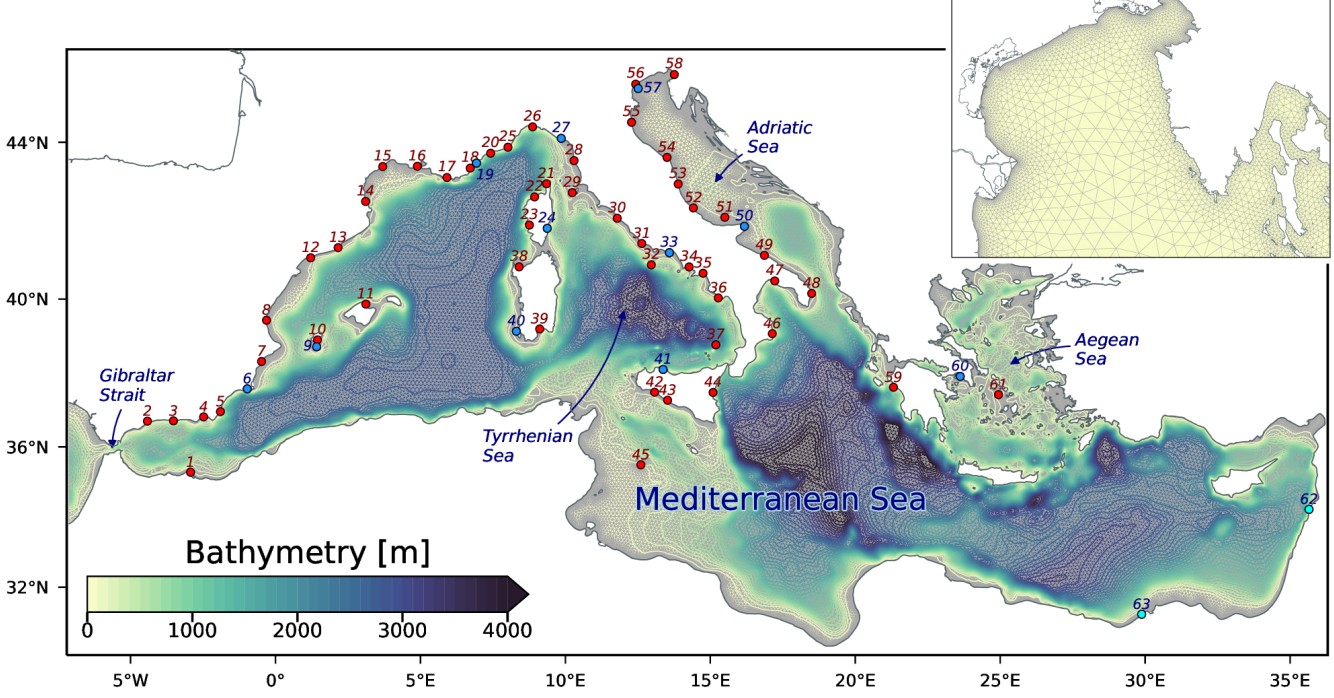

**Figure 1.** The big panel shows the unstructured grid and the bathymetry used by the model. In the small panel a zoom of the grid in the northern Adriatic Sea. The red and blue dots mark the locations of the assimilation and validation tide gauges respectively.

In this application, we use a two-dimensional barotropic formulation given by the following equations:

$$
\begin{aligned}
\frac{dU}{dt} - fV &= -H\left(g\frac{\partial \zeta}{\partial x} + \frac{1}{\rho_w}\frac{\partial p_a}{\partial x}\right) + A_H \nabla^2 U + \frac{1}{\rho_w}\left(\tau_{wx} - \tau_{bx}\right) \\
\frac{dV}{dt} + fU &= -H\left(g\frac{\partial \zeta}{\partial y} + \frac{1}{\rho_w}\frac{\partial p_a}{\partial y}\right) + A_H \nabla^2 V + \frac{1}{\rho_w}\left(\tau_{wy} - \tau_{by}\right) \\
\frac{\partial \zeta}{\partial t} + \frac{\partial U}{\partial x} + \frac{\partial V}{\partial y} &= 0,
\end{aligned}
\tag{1}
$$

where the independent variables are the time, $t$, and the spatial variables $x$ and $y$. $U(x, y, t)$ and $V(x, y, t)$ are the transports along $x$ and $y$, $f(y)$ is the Coriolis coefficient, $H(x, y, t)$ is the sum of the sea depth with $\zeta(x, y, t)$, which is the variable level with respect to the resting state; $g$ is the gravitational acceleration, $\rho_w$ is the average density of seawater, $p_a(x, y, t)$ is the atmospheric pressure at the sea level and $A_H$ is the horizontal coefficient of turbulent viscosity, formulated with Smagorinsky (1963), using a dimensionless coefficient equal to 0.2; while $\nabla^2[\cdot]$ is the two-dimensional Laplacian operator. $\tau_{bx}(x, y, t)$ and $\tau_{bx}(x, y, t)$ are the components of the stress at the bottom, expressed with a linear-quadratic formulation with a coefficient equal to 0.0025 (Bajo et al., 2019); $\tau_{wx}(x, y, t)$ and $\tau_{wx}(x, y, t)$ are the components of wind stress, expressed with the formulation proposed by Hersbach (2011) and with a Charnock coefficient equal to 0.02. Furthermore, for the simulations that calculate

the tidal level or the total sea level, the terms of the tidal potential are also active and four semi-diurnal ($M_2$, $S_2$, $N_2$ and $K_2$) and four diurnal components ($K_1$, $O_1$, $Q_1$ and $P_1$) are calculated.

The model is applied on a mesh of the Mediterranean Sea, which extends into the Atlantic Ocean up to about 7° W and has about 163,000 triangular elements. The size of the elements is variable, with a gradually greater resolution from the open sea (element side length $\sim$ 12 km), to the coasts (element side length $\sim$ 500 m), as shown in Fig. 1. The bathymetry derives from the 2020 dataset of the European Marine Observation and Data Network (https://www.emodnet-bathymetry.eu/), which was bilinearly interpolated on the mesh.

This model has already been used in the past, with similar configurations, in scientific works and is currently used in several operational systems for the sea level prediction. For example, the most extreme storm surge events that occurred in 1966, 2018 and 2019 were studied and simulated in Roland et al. (2009), Cavaleri et al. (2019) and Ferrarin et al. (2021). Various operational versions of the model with similar configurations have been used for over fifteen years at the high tide forecasting and warning centre (CPSM) in Venice (Bajo et al., 2007; Bajo and Umgiesser, 2010; Bajo, 2020) and at the Italian Institute for Environmental Protection and Research (ISPRA - https://www.venezia.isprambiente.it/ispra/modellistica). In this Institute, a system similar to that described in this paper will be installed in the next months. SHYFEM, with an old DA system, was also used to assess the impact of altimeter data on storm surge forecasting (Bajo et al., 2017), and, using the more recent DA system described in this paper, to study a particular seiche event (Bajo et al., 2019). As regards the reproduction of the astronomical tide in the Mediterranean and Black seas, a first specific work has been successfully completed (Ferrarin et al., 2018), but a preliminary total sea level operational system was set up earlier (Ferrarin et al., 2013). Finally, there are numerous works performed with other hydrodynamic models with barotropic configuration for the study and prediction of tides, surges, seiches and sea level variations given by these components (see e.g., Flowerdew et al., 2010; Bertin et al., 2014; Fernández-Montblanc et al., 2019; Horsburgh et al., 2021; Byrne et al., 2021).

### 2.1.1 Surface and lateral boundary conditions and perturbation methods

The simulations use, as surface boundary condition, 10-m wind and mean sea level pressure hourly fields provided by the BOLAM atmospheric model (Mariani et al., 2015), which is hydrostatic and runs at 8 km of horizontal resolution. The model is nested in the ECMWF Integrated Forecasting System (IFS - https://www.ecmwf.int/en/publications/ifs-documentation). In the hindcast/reanalysis simulations, the surface forcing fields are made by the first forecast days chained together, while the forecast simulations, which are daily, use the entire forecast up to five days ahead.

The lateral boundary conditions are closed everywhere except at the western border in the Atlantic Ocean, near Gibraltar, where the sea level is imposed and the water transports are left free to adjust (Dirichlet conditions). The open boundary was chosen outside the Mediterranean Sea to reduce the error inside the basin and different sea level quantities are used, depending on the simulation type. For the simulations computing the total sea level we used the variable *Sea Surface Height (SSH)* by the Mediterranean Sea Physical Analysis and Forecast system (Clementi et al., 2021, https://doi.org/10.25423/CMCC/ MEDSEA_ANALYSISFORECAST_PHY_006_013_EAS7), running at the Copernicus Monitoring Environment Marine Service (CMEMS). For the simulations computing only the surge, we used the "de-tided" SSH, available in the same dataset,

which is the residual part remained after the harmonic analysis of the SSH. Finally, the simulations computing only the tide use the difference between these two quantities. The SSH and the de-tided SSH can contain a baroclinic part which has lower-frequency variations and cannot be easily filtered out.

In the present work, in which the considered domain is relatively small compared to the speed of the barotropic perturbations, the lateral and surface boundary conditions greatly influence the solution of the equations of motion. Therefore, the physical problem can be defined more as "boundary driven" than "initial-state" driven and the perturbation of the surface/lateral boundary conditions is necessary to prevent the narrowing of the initial ensemble after a short time. In all the DA simulations, the members of the ensemble are created by perturbing the initial state and then the spread is maintained by the perturbation of the forcing, the boundary conditions and some model parameters. The perturbation of the initial state is performed only for the sea level (variable $\zeta$ in the eqs. 1), with a technique similar to that used for the atmospheric pressure (described later), while the water transports are not perturbed.

In the forecast simulations, the initial state is perturbed only in the first simulation, then the following daily simulations start from the states saved in the previous-day simulations. Even for the reanalysis simulations, the perturbation of the initial state is not very important, as the simulations last two months and the influence of the forcing and boundary conditions, as well as the assimilated observations, are far more important after some days. Therefore, in reanalysis the forcing and boundary conditions are perturbed for the entire period of the simulations, while, in forecast, each simulation assimilates observations for 24 hours, during which conditions are perturbed, and then five days of deterministic forecast follow, starting from the analysis ensemble mean and using unperturbed forcings and boundary conditions.

The perturbations are calculated so that, for a scalar physical variable, the mean of the perturbed values should be approximately equal to the non-perturbed value and the standard deviation should resemble the estimated error; furthermore, the perturbations must belong to a Gaussian distribution. We used this method for the conditions at the lateral open boundary, with the same perturbations in each node. A similar perturbation was used also for the value of the drag coefficient in the bottom stress, with a distribution centered at 0.0025 and with a standard deviation of 0.0005. In the DA simulations using the tidal forcing (tide and total sea level), a calibration factor for the loading tide (parameter *ltidec* in SHYFEM) is perturbed as well, with a mean value of 6.e-05 and a standard deviation of 1.e-05.

Perturbing the two-dimensional atmospheric fields is more complex. We still impose the same condition for the mean and the standard deviation at each point, but the perturbations must have a spatial correlation and the atmospheric pressure perturbations should be linked to the wind perturbations. We therefore first perturbed the atmospheric pressure field, through a technique to generate two-dimensional pseudo-random fields (Evensen, 1994, 2003), imposing a decorrelation length of about 400 km and a standard deviation of 3.5 hPa. These values, slightly different from those used in Sakov et al. (2012), were found empirically and they produce perturbations at a sub-synoptic scale, with a similar size to the typical Mediterranean cyclones (Ferrarin et al., 2021). From these fields of pressure perturbations, we calculated the corresponding perturbations for the velocity components.

If the pressure perturbation in one point is $\delta P$, the perturbations for the wind components, in geostrophic equilibrium, are:

$$\delta u = -\frac{\delta P}{\delta y} \frac{1}{\rho_a f}$$
$$\delta v = \frac{\delta P}{\delta x} \frac{1}{\rho_a f}. \tag{2}$$

Using these perturbation fields to be applied to the unperturbed fields of wind and pressure at an instant $t$, we obtain perturbed fields with physical coherence. Again for the atmospheric fields, in addition to this kind of perturbation, a temporal perturbation has also been introduced in which, from a field at time $t$, an ensemble of equal fields is generated but with reference time $t+dt_n$, where $dt_n$ are time perturbations belonging to a Gaussian distribution as well.

Finally, as regards the perturbations of the forcing and the boundary conditions that vary over time, the error at a given instant $t_1$ must be correlated to the error at the next instant, $t_2$. This is defined as "red noise" and is implemented by calculating a weight dependent on the time interval between the two fields and by defining a decay time:

$$\alpha = 1 - \frac{t_2 - t_1}{\tau}, \tag{3}$$

where $\tau$ is the decay time. The perturbation $\xi_2$, at time $t_2$, becomes a linear combination of the perturbation $\xi_1$, at time $t_1$, and the newly calculated perturbation $\xi_2^*$:

$$\xi_2 = \alpha \xi_1 + \sqrt{1 - \alpha^2} \xi_2^*. \tag{4}$$

## 2.2 Observations

### 2.2.1 In-situ data

Sea-level observations were retrieved from the European Joint Research Center database (https://data.jrc.ec.europa.eu/). As shown in Fig. 1, tide gauges are concentrated in the western and central Mediterranean Sea, mostly along the Spanish, French and Italian coasts, while on the northern African coast there is only one station (Melilla) and few stations are present in the eastern Mediterranean Sea. The Adriatic Sea has stations only along the Italian coast and not on the eastern coast, but they are still quite numerous. The stations in the Mediterranean Sea were divided into 50 stations to be assimilated and 13 stations for the validation (Tab. A1). The data are recorded every 10 minutes in the period of October-December 2019. We processed them with the SELENE quality check software (https://puertos-del-estado-medio-fisico.github.io/SELENE/; Pérez et al., 2013) for spikes and outliers detection, stability test, date and time control, flagging and interpolation of short gaps. Subsequently, the quality-checked data were elaborated with the Python binding of UTide (https://github.com/wesleybowman/UTide; http://www.po.gso.uri.edu/~codiga/utide/utide.htm), based on the least squares fitting, to separate the tidal periodic part from the Non-Tidal Residual (NTR). We kept the eight most energetic tidal constituents in the harmonic analysis ($M_2$, $S_2$, $N_2$, $K_2$, $K_1$, $O_1$, $P_1$, $Q_1$), which are the most important in the Mediterranean Sea (Ferrarin et al., 2018). The NTR was further processed by applying a 2-hour moving average, to remove high-frequency signals. The harmonic analysis was not possible for stations 62 and 63, due to the lack of enough continuous data. Therefore, these stations were used only for the validation of the total sea level for which the harmonic analysis is not necessary.

Finally, the observations from different stations have often different mean sea levels. Sometimes this is due to different reference datum, which depends on the monitoring network they belong. Furthermore, the observed sea level can contain a low-frequency non-barotropic part due to salt and temperature gradients, as well as steric effects. Therefore, we decided to refer all the observations to the two-month mean sea level computed by a deterministic simulation of the model that we used. A similar approach is adopted in Byrne et al. (2021).

### 2.2.2 Altimeter data

Altimeter data are difficult to use in storm surge studies, even if some attempts were made in the past (Bajo et al., 2017). Since high-frequency signals are badly sampled by the satellite tracks, usually this part is removed with the help of a barotropic two-dimensional model (Carrère and Lyard, 2003). Normally, in the altimeter products, also the tidal part is removed with a similar barotropic tidal model (Lyard et al., 2021). However, since the altimeters measure the sea level every cycle (about 10 days) in the same locations, it is possible to extract the tidal part from the signal by means of harmonic analysis.

Recently, the amplitudes and phases of the main harmonic components along the altimeter tracks have become available on the AVISO website (https://doi.org/10.6096/CTOH_X-TRACK_Tidal_2018_01). The X-TRACK along-track tidal constants were computed via harmonic analysis of the sea level anomalies for long time series missions (Birol et al., 2017). We used the X-TRACK (based on Topex/Poseidon + Jason-1 + Jason-2) eight most energetic tidal constituents over the Mediterranean Sea (see the list in the previous section) to compute the astronomical tide for the period of our simulations. This data was used in the validation of the tidal reanalysis simulation, as described in Section 3.2.

### 2.3 The data assimilation system

In this section and the following ones, we will use some terminologies and concepts typical of the DA, for an introduction to these concepts and to the different techniques we recommend reading Carrassi et al. (2018).

The code used for the DA in this paper is based on routines developed and described in Evensen (2003, 2004) and available at https://github.com/geirev/EnKF_analysis. These routines have been adapted and extended to be used in the SHYFEM model, allowing different techniques, such as the Ensemble Kalman Filter (EnKF) and the Ensemble Square Root Filter (EnSRF), and the use of different numerical schemes (https://github.com/marcobj/shyfem). Furthermore, various routines have been created to perturb the forcings and boundary conditions in order to obtain ensembles of arbitrary size. In the present work, we used the EnKF with the correction described in Evensen (2004) to avoid the loss of rank in the observation covariance matrix (Kepert, 2004). The system uses the adaptive inflation (Evensen, 2009a) to avoid narrowing of the ensemble spread; while the observations are considered independent (they come from different stations). Therefore, the observation covariances are set to zero, while the variances are positive and equal to each other. In order to discard the too high innovations, a simple technique checks the values of the variances of the background matrices and the observations (Järvinen and Undén, 1997; Storto, 2016).

Finally, to avoid shocks in the model state near the lateral open boundary due to the imposition of the sea level, in the final ensemble states the analysis states are relaxed to the background ones, gradually approaching the boundary. The background

and analysis states are weighted through a Gaspari-Cohn (GC) function (Gaspari and Cohn, 1999), prescribing a radius from the nodes of the lateral open boundary. In each node the model ensemble states after an analysis step is:

$$A_a^*(x,y) = A_b(x,y)f(x,y) + (1 - f(x,y))A_a(x,y),$$ (5)

where $x$ and $y$ define the position of the node in the grid, $A_b$ are the background states, $A_a$ are the analysis states, $f$ is the GC function, equal to 1 in the open boundary nodes. Since the GC function goes to zero at a distance greater than twice the radius, after this distance the solution is identical to that of the analysis, while near the boundary is similar to the background solution, strongly driven by the boundary condition and it is not affected by the analysis increments.

This set of the DA parameters were decided after running several calibration tests, some of which are exposed in Section 2.4.

## 2.4 Results' production and post-processing

All the simulations were run in the period from the beginning of November to the end of December 2019. The hindcast/re-analysis simulations are two-month long with continuous forcing and boundary conditions, as described in Section 2.1.1. The reanalysis simulations assimilate the data from the 50 stations every hour, throughout the two months. From the ensemble states, the analysis ensemble mean is calculated, as the best estimate of the real state of the physical system, and is used in the examination of the results.

In running the forecast simulations we used the same settings as those that would be used in an operational context. The period is the same as considered in the hindcast and reanalysis simulations. However, the simulations are performed daily and each simulation is composed of a hindcast (no DA) or analysis (DA) simulation of one day and a five-day forecast simulation. For the sake of brevity, we will show only the results of the first three days. The forecast simulations with DA assimilate the data from the 50 stations, every hour, in the 24 hours preceding the forecast. From the final analysis states we computed the analysis ensemble mean, each day at 00 UTC, and we used it as initial state to run five-day forecast. Then, the analysis states are saved to be used as initial states in the next day's simulation. In this way, the DA always starts from analysis states and is similar to the cycle performed in reanalysis, except for the perturbation of the forcing and boundary conditions, which is made again every day.

To evaluate the results, each daily forecast simulation was divided into five parts and each part was chained with the corresponding one of the previous and following days. Continuous results are obtained for 1-day, 2-day and 3-day lead times and can be directly compared with the observations. The forecast timeline is shown in Fig. 2 and is the same for the simulations without and with DA.

We calculated the standard deviations of the model and observed data, the correlation between them and the Centered Root Mean Squared Error (CRMSE). The standard deviations and CRMSEs were normalised to the standard deviation of the observations at each station and represented by Taylor diagrams (Taylor, 2001). Bias error plots were also calculated and the bias is calculated as the mean of the differences between the modelled and observed values; while the CRMSE represented in the same plots is not normalised. For the sake of clarity, we reported the various simulations in Tab. 1 with identification labels, which we will use in the following sections.

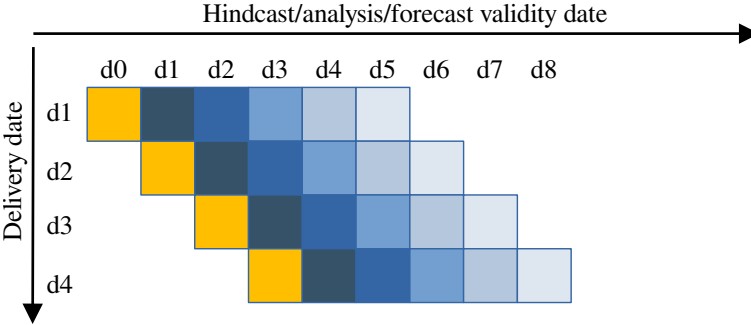

**Figure 2.** Timeline of the forecast simulations. The squares represent the days, which are expressed as d0, d1, etc. The delivery date is the day when the forecast is supposed to be executed, while the validity date is the length of the forecast. The orange squares are the days of hindcast (without DA) or of analysis (DA). The blue squares are the forecast days, from the first (darkest) to the fifth (lightest).

**Table 1.** Clusters of simulations executed in this work. The IDentification label is composed by the physical variable (T - tide, S - surge, Z - total sea level), by the type of simulation (hindcast/reanalysis/forecast) and by the use of DA.

| ID | Variable | Type | DA |
|----|----------|------|----|
| $TH$ | tide | hindcast | no |
| $TR_A$ | tide | reanalysis | yes |
| $SH$ | surge | hindcast | no |
| $SR_A$ | surge | reanalysis | yes |
| $ZH$ | total sea level | hindcast | no |
| $ZR_A$ | total sea level | reanalysis | yes |
| $SF$ | surge | forecast | no |
| $SF_A$ | surge | forecast | yes |
| $ZF$ | total sea level | forecast | no |
| $ZF_A$ | total sea level | forecast | yes |

Regarding the spectral analysis, we used the NTR and the model surge signal in December 2019. The power spectral density was estimated with the Welch method (Welch, 1967), dividing the period into 8-day windows with 50% overlap. The fast Fourier transform length is rounded up to the nearest integer power of 2 by zero padding.

## 3 Results

### 3.1 Calibration of the data assimilation

Before running the final simulations used to produce the results, we carried out numerous experiments to determine the best values of some DA parameters. The parameters that have been varied are the assimilation scheme (EnKF, EnSRF), the error

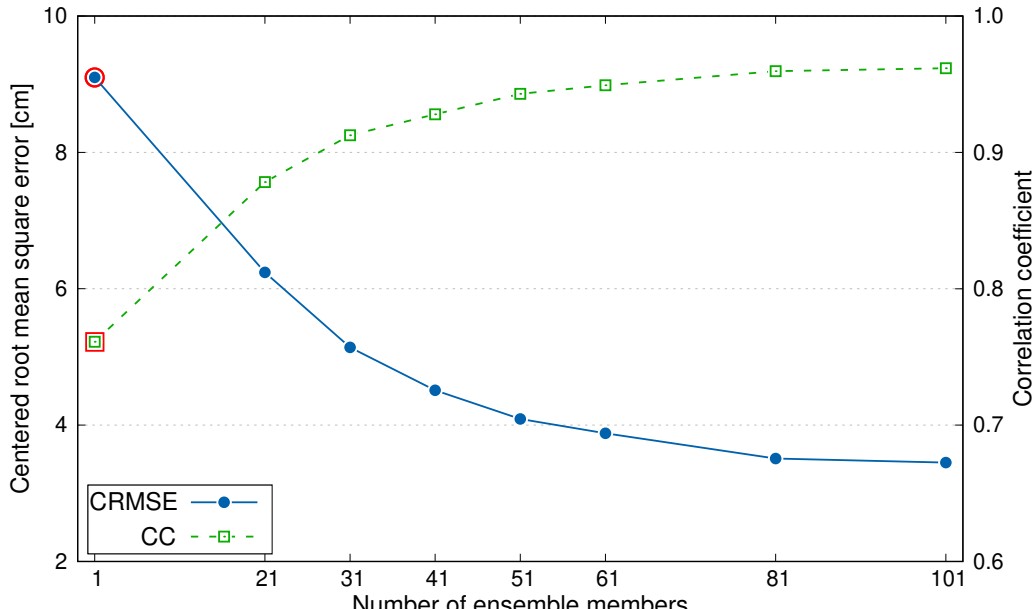

**Figure 3.** Performance of the data assimilation, in terms of CRMSE and correlation coefficient, as a function of the number of ensemble members. The red contour highlights the results of the simulation without data assimilation.

of the observations (we tested from 1 cm to 3 cm), the radius in eq. 5, the radius in the domain localisation and the number
of the ensemble members. Although in fact, the localisation brings advantages in many applications, in our case the available observations are mainly located in the northern side of the computational domain. This implies that to obtain a spatially uniform analysis correction, a large localisation radius should be used to reach the other border of the basin. Furthermore, the correlation radius of a variable (barotropic sea level perturbations in our case) between a point and its neighbours increases with its propagation speed. In the present case, the propagation speed is that of shallow water waves (in the western Mediterranean
basin, considering an average depth of about 2000 m, the speed is 140 m/s). For these considerations and after having carried out various tests varying the radius of the local analysis, we have decided not to use it and to increase the number of ensemble members. A high number of ensemble members avoids problems of spurious correlations and cross-correlations. Moreover, since the simulations are extremely fast and having a workstation with a high number of cores, the execution time has not been much affected. To determine the minimum number of ensemble members to obtain good results without increasing too
much the computational load, we performed various total sea level reanalysis simulations. In Fig. 3 we report the Centred Root Mean Squared Error (CRMSE) of the analysis ensemble mean, averaged in the validation stations, using a different number of ensemble members. The error is reduced from 9.3 cm, in the case without DA, to 3.6 cm using 101 members, and the correlation increases from 0.75 to 0.95. Since the error pattern is regular and asymptotic, we decided to use 81 members.

     Therefore to conclude, the final configuration uses the EnKF with an observation error of 2 cm, a radius in eq. 5 of 250 km,
no localisation techniques and 81 ensemble members.

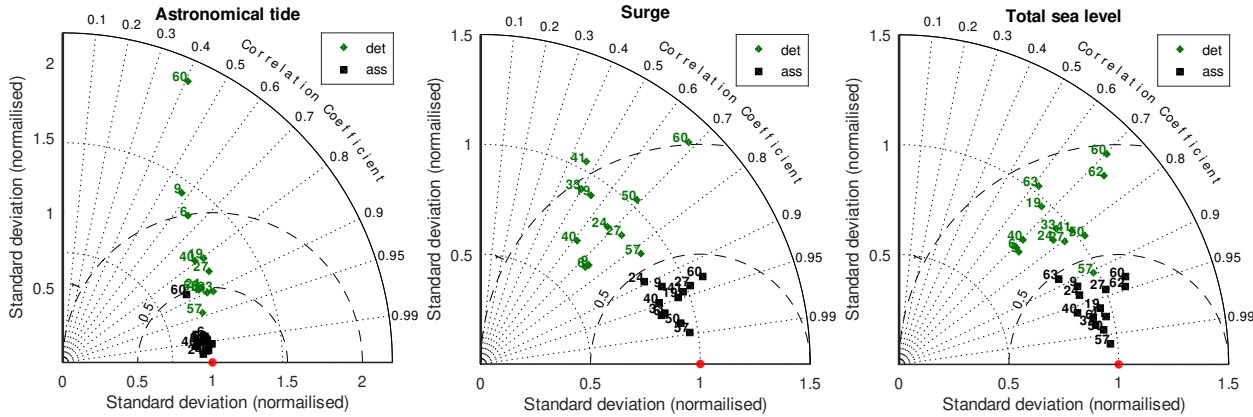

**Figure 4.** Normalised Taylor diagrams of the hindcast and reanalysis simulations. The deterministic simulations (green diamonds) compared to DA simulations (black squares), for the astronomical tide (left), the surge (centre) and the total sea level (right). The red dot indicates the perfect agreement.

### 3.2   Hindcast/reanalysis simulations

In this section we analyse the results of the hindcast and reanalysis simulations, for the astronomical tide, the surge and the total sea level. In Fig. 4, the first diagram on the left shows the astronomical tide comparison, in which the model results, without (hindcast) and with (reanalysis) DA, are compared with the tide calculated by the harmonic constants ($TH$, $TR_A$). The results

are good even without DA in almost all stations, with a certain tendency to overestimate the signal amplitude (higher standard deviation). Station 60 is an exception, where the results in hindcast are poor, probably due to its position in the Aegean Sea, a morphologically complex area. The results with DA are very good for all the validation stations, reaching almost perfect agreement (correlation about 0.99), with a small deterioration in station 60, which however improves and still achieves a more than good accuracy (CRMSE from 4 cm to 1 cm).

The central diagram shows the reproduction of the surge signal, compared with the NTR extracted from the observations ($SH$, $SR_A$). In this case, the distribution of the stations in the Taylor diagram is sparse for the deterministic simulation and the station 60 is still the worst. The reanalysis simulation improves considerably the surge reproduction in all the stations, with a very focused distribution even if not like that of the astronomical tide. For example, in station 60, the CRMSE reduced from 8 cm to 3 cm.

Finally, the simulations with the total sea level ($ZH$, $ZR_A$) have a quality similar to that of the surge simulations. Some stations are even better, perhaps thanks to the good accuracy in the reproduction of the tidal signal. As for the surge simulations, the CRMSE goes from 8 cm in the hindcast simulation to 3 cm in the reanalysis.

For the total sea level, we made a comparison also for the stations 62 and 63 which, as previously mentioned, are the only ones in the eastern basin and are at least a thousand kilometres away from the nearest assimilated station. It is interesting to

note that these stations have a consistent improvement; the CRMSE goes from 9.6 cm to 4 cm for station 62 and from 10.9 cm

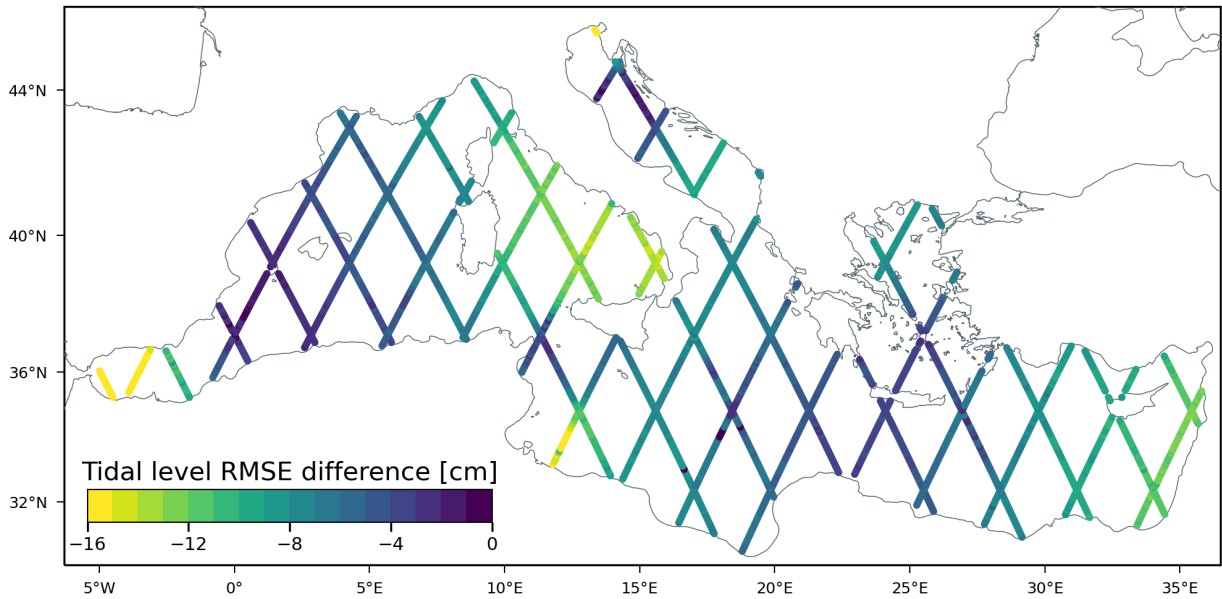

**Figure 5.** CRMSE differences (ass - det) for the tidal level computed using the altimeter X-TRACK along-track tidal constants retrieved from AVISO.

to 5.7 cm for station 63. Probably, this improvement is due to the high number of ensemble members, which allows correct correlations in the background covariance matrix, even for variables very distant from each other.

In order to validate the DA even in the open sea, far from the coasts, it is possible to use altimeter data for the computation of the harmonic constants and the tide. The amplitudes and phases of the eight most energetic tidal constants retrieved from the altimetric data were used to calculate the tide oscillations at each point of the satellite tracks in the Mediterranean Sea. To compare this data with the model data, the sea levels from the $TH$ and $TR_A$ simulations were extracted at the same coordinates and the CRMSE were calculated. Fig. 5 shows the along-track differences in the CRMSE (i.e., CRMSE$_{TR_A}$ - CRMSE$_{TH}$). The values are negative almost everywhere, clearly showing a marked improvement of the DA in reproducing the tidal levels over the whole basin with a reduction of the CRMSE up to 20 cm near the Gibraltar Strait, in the Gulf of Gabes and in the northern Adriatic Sea. It is worth noting that, the DA effect is not local, as the areas in which there is a greater improvement do not correspond totally to those with more assimilated stations (e.g., the eastern Mediterranean Sea). Averaging the CRMSE over the whole basin, we obtain a mean value of 11.6 cm for the simulation without DA ($TH$) and a value of 4.3 cm for the simulation with DA ($TR_A$).

### 3.3 Forecast simulations

In this section we analyse the results of the forecast simulations for the surge component and for the total sea level. In Fig. 6 the Taylor diagrams show the comparison with the observations for the first, second and third forecast days, both for the model

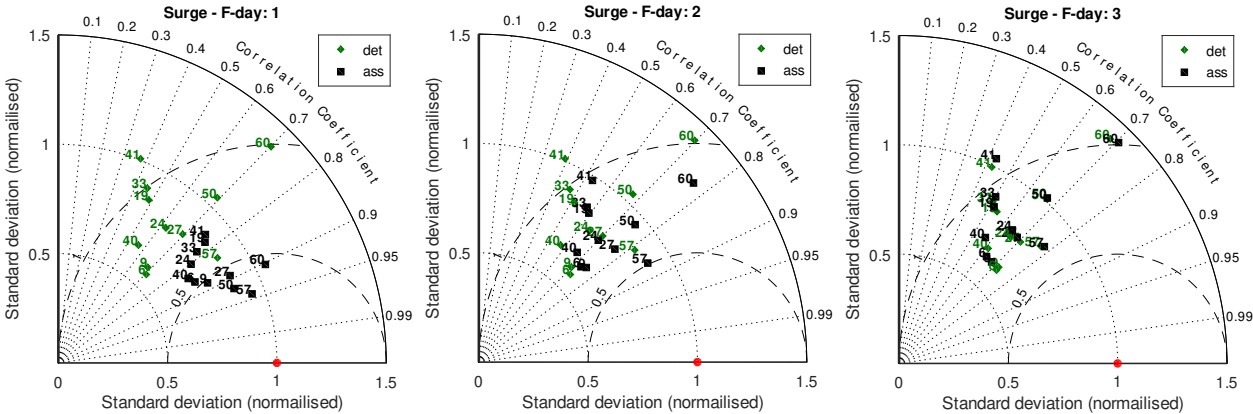

**Figure 6.** Normalised Taylor diagrams of the forecast simulations with the surge. The deterministic simulations (green diamonds) compared to DA simulations (black squares), for the first (left), second (centre) and third (right) -day forecast. The red dot indicates the perfect agreement.

data without DA, starting from a background state and for those with DA, starting from the analysis ensemble mean. In the results relative to the surge simulations ($SF$, $SF_A$), the effect of the DA on the first forecast day is evident and the distribution is similar, slightly worse, to that obtained in the hindcast and reanalysis simulations in Fig. 4, central panel. The data improves

in each validation station, including station 60, which is far from the nearest assimilated station. Unfortunately, the data in stations 61 and 62 cannot be used in the validation of the surge simulations, as it was not possible to perform the harmonic analysis necessary to subtract the tide, due to the scarcity of available data.

The improvement is smaller on the second day forecast, while on the third day is almost nil, worsening slightly in some stations. This behaviour is due to the fact that the initial state of the system gradually loses its importance as the forecast moves

away from it, as well as the error correction. The forecast without DA has a larger error in the initial state, which mostly counts on the first and second days of forecast.

In Fig. 7 we show the bias error for the surge simulations. This plot was not made for the hindcast and reanalysis simulations for which the bias is almost null. The figure shows that the DA improves the results, especially on the first forecast day, then the correction is still positive but weaker on the second day, while on the third day the DA slightly worsens the original forecast,

in agreement with what has been seen in the Taylor diagrams. The worsening is contained and relates to the third forecast day which, in an operational context, is of secondary importance compared to the first and second days. Still, observing Fig. 7, it can be seen how station 57 deviates from the others, with a much greater bias and CRMSE. This is due to the position of this station, in the northern Adriatic, where the surge signals and the associated seiche oscillations are larger than in the rest of the Mediterranean Sea. However, precisely for this reason and since there are numerous good-quality stations in the Adriatic

Sea, the effect of DA is strong, both in the correction of random and systematic errors. The systematic errors, represented by the biases in Fig. 7, are almost all positive, denoting an overestimation of the model. This behaviour is true for the two-month period considered, while for extreme events the trend is normally the opposite.

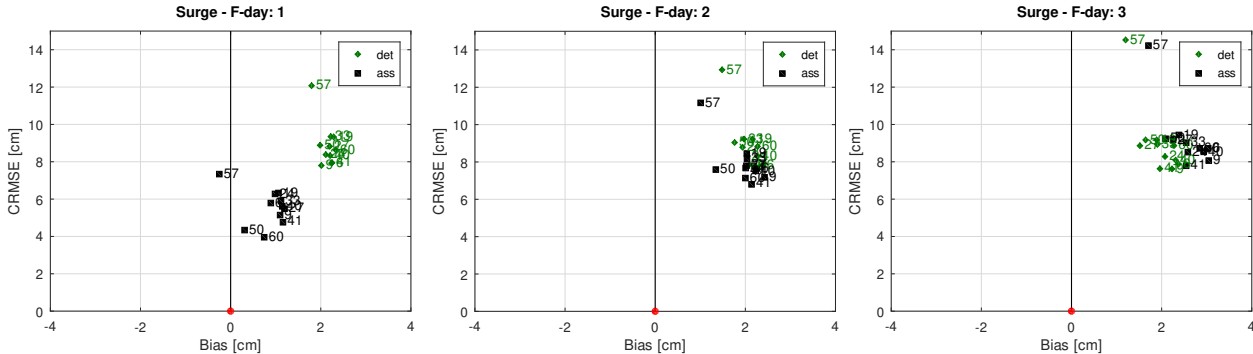

**Figure 7.** Bias diagrams for the first (left), the second (centre) and the third (right) -day forecast of the surge simulations. The deterministic results (green diamonds) are plotted with the DA ones (black squares). The red dot is the perfect agreement, while positive bias means an overestimation of the model.

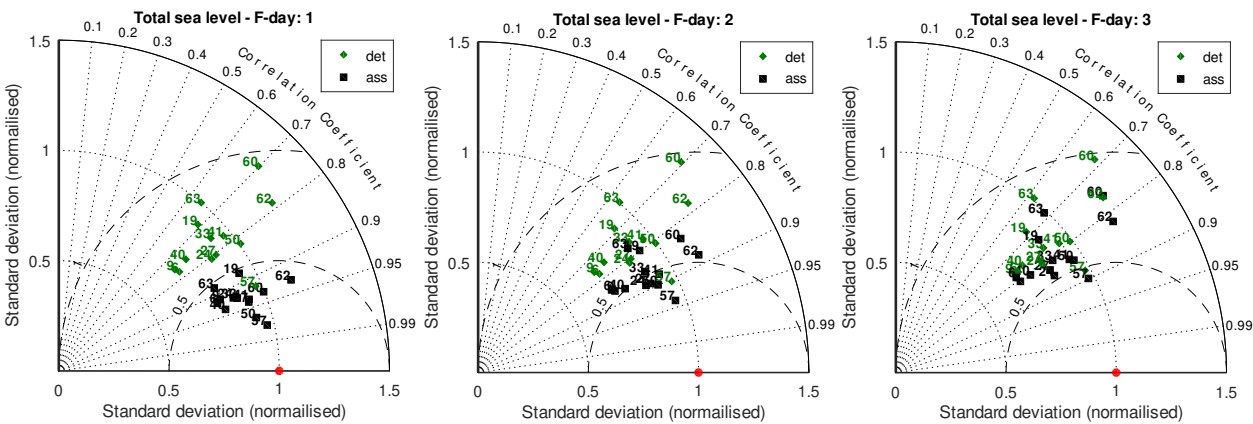

**Figure 8.** Normalised Taylor diagrams of the forecast simulations with the total sea level. The deterministic simulations (green diamonds) compared to the DA simulations (black squares), for the first (left), second (centre) and third (right) -day forecast. The red dot is the perfect agreement.

In Fig. 8 we report the Taylor diagrams for the total sea level ($ZF$, $ZF_A$). In this case, the results are slightly better than for the surge. The simulations, both without and with DA, maintain evident improvements even on the third forecast day. For the total sea level, we can evaluate the improvement also in the stations 61 and 62, even if they have a smaller number of records. As seen for the hindcast/reanalysis simulations, these stations are important because of their distance from other assimilated stations and because they are the only stations in the eastern Mediterranean basin. In these two stations, the DA improves strongly the results as well as in the reanalysis simulation.

Fig. 9 shows the bias diagram for the total sea level. As for the surge, the biases are positive in most of the stations, denoting a model overestimation, but they are generally lower than those of the surge, even for the model without DA. As shown in the Taylor diagrams, also the CRMSEs in Fig. 9 improve with the DA in all the three days of forecast, even if more in the first.

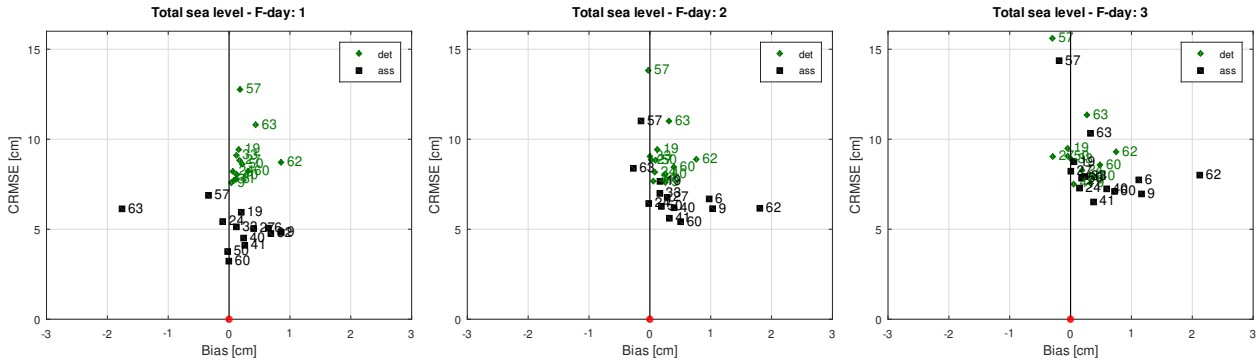

**Figure 9.** Bias diagrams for the first (left), second (centre) and third (right) -day forecast of the total sea level simulations. The deterministic results (green diamonds) are plotted with the DA ones (black squares). The red dot is the perfect agreement, while positive bias means an overestimation of the model.

### 3.3.1 12 November 2019's storm surge event

On 12 November 2019, a particularly intense meteorological perturbation hit the central part of the Mediterranean basin. A sub-synoptic cyclone, centred in the Tyrrhenian Sea, caused a strong south-easterly (Sirocco) wind along the entire Adriatic
basin, with a fairly typical configuration. However, embedded in the first cyclone, a second meso-beta scale cyclone developed near the south-eastern coasts of Italy and moved in the north-westward direction over the Adriatic Sea. This second cyclone moved at a speed close to that of shallow water waves in the northern Adriatic basin, causing Proudman resonance (Proudman, 1929; Ferrarin et al., 2021). In Venice, the sum of the various sea level contributions produced a maximum which was the second highest ever recorded (Ferrarin et al., 2021).
In Fig. 10, we report the sea level forecast, without and with DA, the day before the main peak, the same day and the day after. The sea level is related to the Venice station and the forecasts are retrieved from the simulations $SF$ and $SF_A$ with the addition of the tide computed by the harmonic constants. The previous day's atmospheric forecast underestimated the wind and had strong errors in positioning the cyclones. Consequently, also the sea level forecast had large errors (left panel) and the use of the DA has no effect since the initial state was relative to an instant of calm conditions and did not contain any large
errors. The second forecast, shown in Fig. 10 central panel, is relative to the day of the event. The meteorological forecast was accurate, with a good reproduction of the track followed by the smaller cyclone. Consequently, the prediction of the sea level is good even without the use of the DA since, even in this case, the event started to evolve after the time of the initial state. The DA does not improve the main peak but it corrects slightly the previous peak.

Finally, we show the forecast of the day after because a large peak, even if less extreme than the previous one, was registered
in Venice. This event happened with calm weather conditions and was due to an overlap of the tidal peak to a small seiche peak, probably linked to the second mode of the Adriatic basin (A2 in Tab. 2). The forecast without DA missed the reproduction of this peak because of errors in the initial state of the surge field in the Adriatic Sea. In this case, the DA can give a valuable contribution, with a correction of about 15 cm, which is considerable (Fig. 10, third panel).

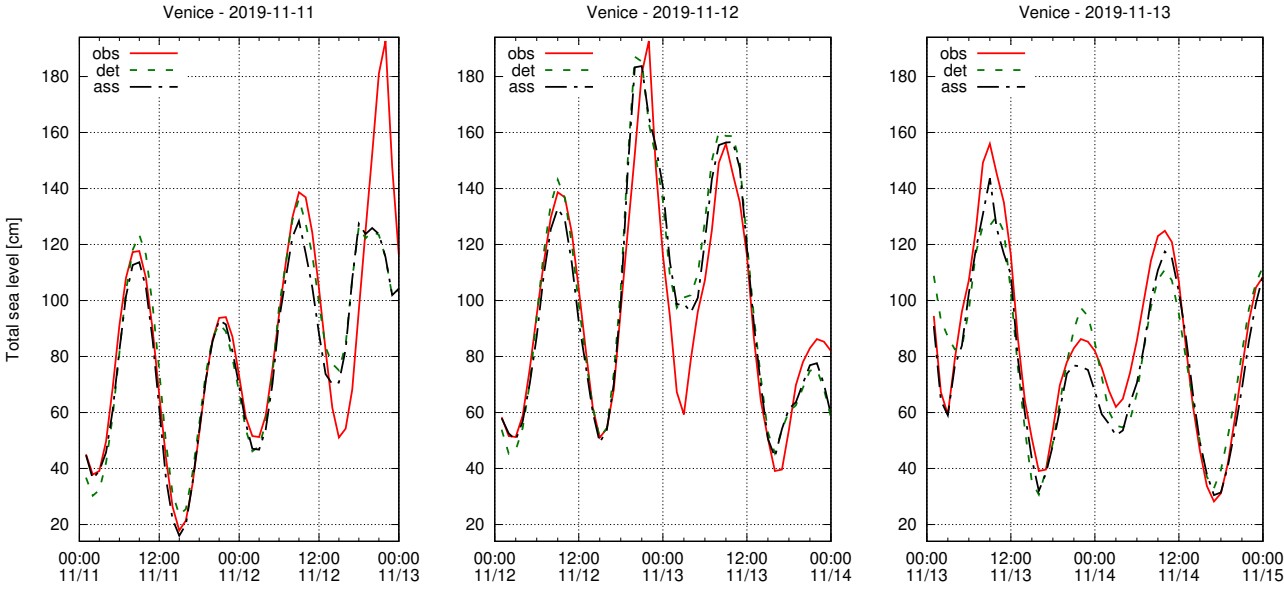

**Figure 10.** Forecasts issued on 11, 12 and 13 November 2019 at the Venice station, from the surge simulations (adding the tide). The observed total sea level (obs) is compared to the forecast without (det) and with (ass) the use of the DA. The sea levels are in CET time and are referred to the local datum (ZMPS).

### 3.3.2 December 2019's seiche events

As explained in the introduction, seiches are free barotropic oscillations of the sea level in a basin, triggered by an initial perturbation. Therefore, since they are not forced, the reproduction of their propagation depends solely on a correct initial state and a correct modelling setup. Given that DA has the purpose of reducing the error of the initial state, we expect, as shown in the previous section, a remarkable impact on the reproduction of the seiches.

In December 2019 (period included in our simulations), significant seiche events, among the most energetic ever recorded in
this area, took place (Fig. 11). Despite their intensity, they were not preceded by any strong storm surge. A possible explanation could be that these oscillations were triggered by a slightly-periodic atmospheric oscillation at a frequency similar to that of the normal modes of the basin (which have the basin's resonant frequencies).

These events were poorly predicted by storm surge models operating at that time in Venice (none with DA), the city most affected by flooding in the northern Adriatic. Fig. 12 shows the total sea level recorded in station 56 (Venice) and the first three
days of forecast from the surge simulations ($SF$, $SF_A$ with the addition of the astronomical tide). The oscillations observed in the figure are therefore a superposition of the astronomical tide on the surge signal, which is dominated by the seiche oscillation. At the beginning of the forecast, the DA corrects an error of about 30 cm and maintains a continuous improvement over time, which can also be appreciated after three days of forecast. Although in the section 3.3 we have seen that the statistical

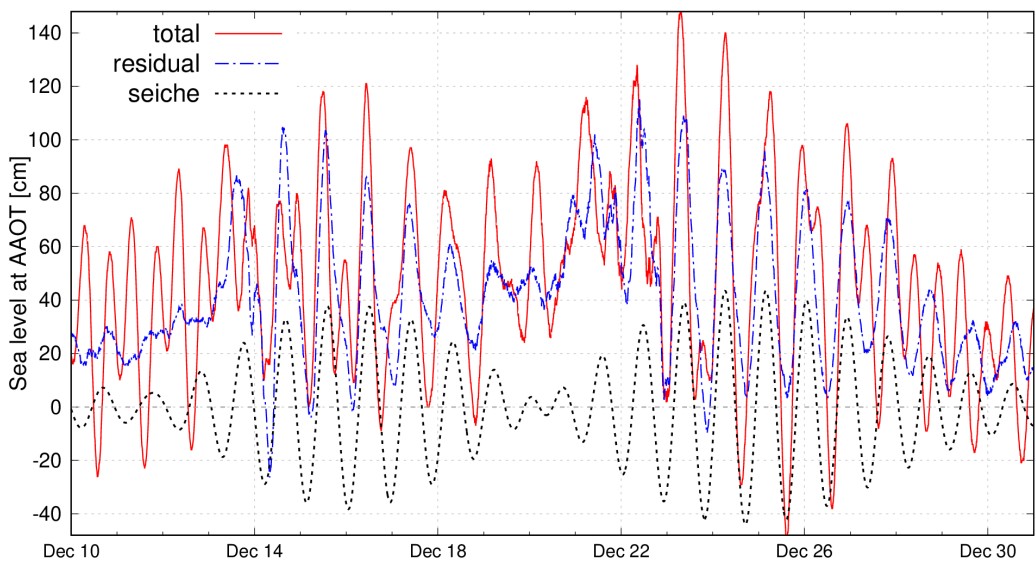

**Figure 11.** Seiche event happened on December 2019, recorded at the AAOT station (n. 57). From the observed total sea level (total) we extracted the NTR (residual) and the seiche contribution (seiche), with a bandpass filter. The sea levels are in CET time and are referred to the local datum (ZMPS).

improvement at three days is not very appreciable, when these oscillations are considerable the error of the initial state tends
to be larger and the DA provides a greater correction.

To check the spatial patterns of the DA correction in this event, we plotted in Fig. 13 the surge increments of the analysis ensemble mean with respect to the background ensemble mean, averaged over one daily DA cycle, on 14 December 2019. The increments are distributed equally throughout the domain and do not appear to be concentrated in the areas with more stations. This is correct as variations of barotropic phenomena, which have a very large spatial scale, must be extensive. There could
be some wrong increments in the southern and eastern areas, where no stations are assimilated, however, this does not seem to emerge from the statistic of the results, which is good also in this part (e.g., station 60 in Fig. 6 and stations 60, 62, 63 in Fig. 8). Finally, note how the increments tend to zero near the open boundary in the Atlantic, as a consequence of the eq. 5, to avoid shocks given by the prescribed sea level.

These events demonstrate the particular effectiveness of the DA in correcting the dynamical state in presence of seiches. To
better highlight this feature we carried out the spectral analyses of the NTR from the observations and the model surge, in all the stations for December 2019. From the peaks in the power spectra, the periods and energy of the excited barotropic modes can be deduced. Before examining the model performances in the reproduction of the power spectra, we give below a summary of the periods of the barotropic modes of the Adriatic and Mediterranean seas. We report the values currently known from works based on observations or models and the periods extracted from our observations (see Tab. 2). Although the periods of
the main modes of the Adriatic Sea are known (Cerovecki et al., 1997; Vilibić et al., 2005; Vilibić, 2006; Bajo et al., 2019),

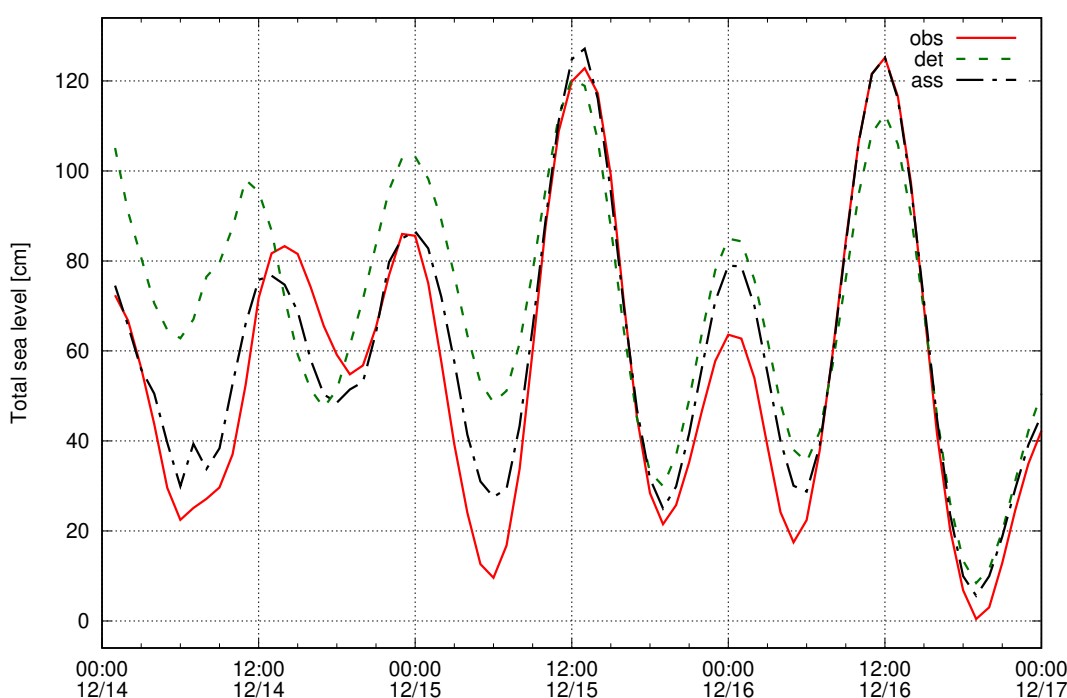

**Figure 12.** Forecast issued on 14 December 2019 at the Venice station, from the surge simulations (adding the tide), referred to the local datum (ZMPS) and in CET time. The observed total sea level (obs) compared to the forecast without (det) and with (ass) DA.

no works based on the analysis of observations (as far as we know) report the periods of the Mediterranean basin. For the Mediterranean Sea we found only a work that reports the shapes and the periods of the main modes, deduced with a simple model, with remarkable accuracy (Schwab and Rao, 1983). In this work, the authors calculated the eigenvalues of a simplified barotropic model of the Mediterranean Sea and found four modes of oscillation. The first mode (M1) relates to an oscillation with a single positive amphidromic node in the Gulf of Sicily and an expected period of 38.5 hours. This mode, which should have maximum amplitude both at the western and eastern borders of the Mediterranean basin, has not been identified by our observations, probably because it has not been solicited by any forcing in the period that we considered.

The second mode (M2) has a more complex shape with a negative amphidromic node in the western basin, a positive one in the eastern basin and a third one in the Adriatic. This oscillation has an expected period of 11.4 hours. A similar peak, with a period of 12.8 hours, is present in the power spectra of several stations of the western basin (Fig. 15). The difference from the expected peak can be explained by considering the various simplifications and the low resolution of the model used in Schwab and Rao (1983), which dates back many years ago.

The third mode (M3) has three positive amphidromic nodes in the Mediterranean basin and one positive and one negative node in the Adriatic basin. This mode has a period of 8.4 hours and maximum amplitude near the Gibraltar strait and along the west coast of the Adriatic Sea. Indeed, from our measurements, a peak at 8-8.3 hours is quite evident in some stations in the

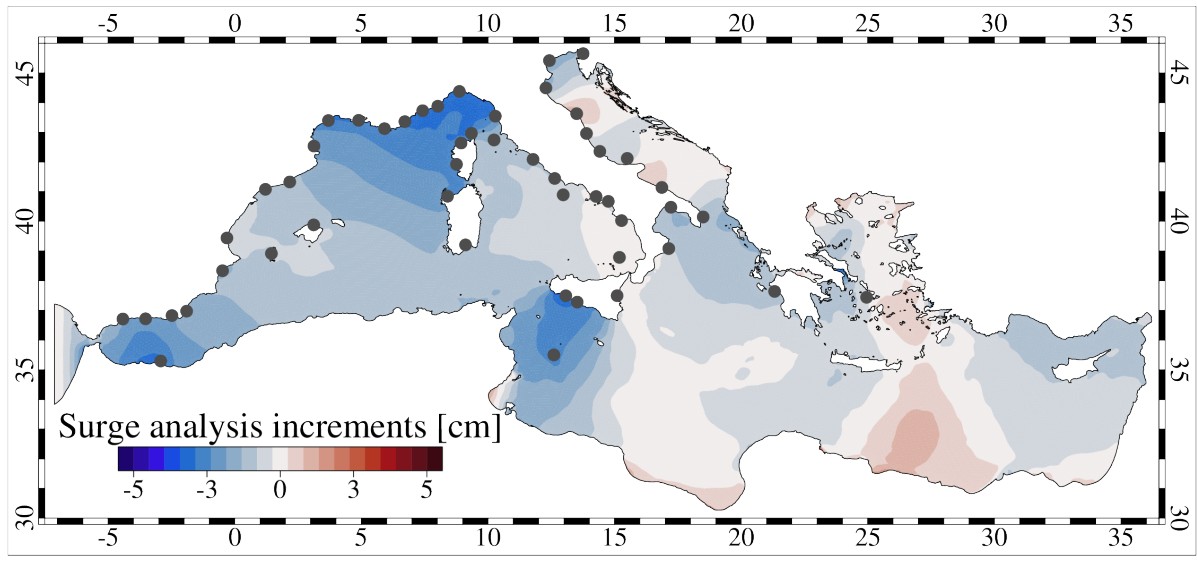

**Figure 13.** Surge increments of the analysis ensemble mean with respect to the background ensemble mean. The increments are the mean of 24 hourly steps (one daily DA cycle) on 14 December 2019. The assimilated stations are marked as dark-grey dots.

western Mediterranean basin (Fig. 15) and a hinted peak is also present in Trieste (Fig. 14) and in other stations on the western coast of the Adriatic Sea.

Finally, the fourth Mediterranean mode (M4) of 7.4 hours should be related to the main oscillation of the Tunisian bight, where we have no observations and therefore we cannot check its presence. From the observation power spectra that we have

analysed, there seems to exist a fifth mode, that we called M5, visible in the stations of the western Mediterranean basin and with a period of 6.2 hours (Fig. 15). However, we have no information of this oscillation from the scientific literature of our knowledge.

Regarding the Adriatic Sea, the fundamental mode, here referred to as A1, is an oscillation that covers the entire basin, with a nodal line south of the Strait of Otranto, near the 1000 m bathymetric line, and has a period of about 21.2 hours. This

oscillation is the most energetic among those analysed and is clearly visible in the observation power spectra, with a period of 21.3 hours (Fig. 14).

The second Adriatic mode (A2) has a nodal line that cuts the basin north of Ancona and a second line south of the nodal line of the fundamental mode, near the 2000 m bathymetric line. This oscillation is quite energetic, albeit less than the main one, and has a period of about 10.7 hours, which is perfectly confirmed by our observations (Fig. 14). Finally, the third Adriatic

mode (A3) has a nodal line under the Po delta, one just above the Gargano peninsula and a third line coinciding with that of the fundamental mode. This oscillation has a period of about 6.7 hours, but we did not detect it in our power spectra. Probably, even this mode was not triggered during the two-month period that we analysed.

**Table 2.** Periods of the barotropic modes in the Adriatic and Mediterranean basins. A mode-identification label is written in the first column. The second column shows the average periods estimated by scientific works by observation spectral analysis, the third column shows the periods estimated by the model in Schwab and Rao (1983) and the last column shows our estimation of the periods by spectral analysis of the observations.

| Mode ID | $T_{ol}[h]$ | $T_s[h]$ | $T_{op}[h]$ |
|---------|-------------|----------|-------------|
| A1 | 21.2 | 20.1 | **21.3** |
| A2 | 10.7 | 9.3 | **10.7** |
| A3 | 6.7 | 6.8 | - |
| A4 | 5.3 | - | **5.2** |
| M1 | - | 38.5 | - |
| M2 | - | 11.4 | **12.8** |
| M3 | - | 8.4 | **8.3** |
| M4 | - | 7.4 | - |
| *M5* | - | - | **6.2** |

Finally, in Trieste and in other Adriatic stations, there is a peak at 5.2 hours, which we called A4. This peak cannot be the Trieste bay seiche, which has a period of 2.7-4.2 hours (Šepić et al., 2022), and was found also by Šepić et al. (2022), with a value of 5.3 hours. Its origin is still unclear.

After this description of the barotropic modes of the Mediterranean and the Adriatic basins, we show now how the model reproduces them in the first day of the forecast simulations ($SF$, $SF_A$). Fig. 14 shows the power spectra for two stations in the Adriatic Sea, Trieste, in the northern part, and Bari near the end of the basin in the southern part. Both the peaks of the fundamental mode, A1 and that of the second mode, A2, are clearly visible in these stations. Note that the peaks are much more energetic in Trieste than in Bari, which is located near the nodal lines of the two modes. The two peaks are both underestimated by the model without DA, while with the DA the peak of the first mode is reproduced very well, especially in the north. The A2 peak remains slightly underestimated at both stations but improves significantly with respect to the simulation without DA. Finally, in the Trieste station, a peak corresponding to the period of the third mode of the Mediterranean Sea (M3) is slightly visible in the observations. However, the model power spectra, both with and without DA, are noisy in this part of frequencies and do not reproduce it. Still in Fig. 14, but only in the Trieste station, the A4 peak is well visible in the observation power spectrum but it is not reproduced by the model. This peak could be related to some local atmospheric phenomenon not present in our forcing.

In Fig. 15 we show the power spectra of two stations near Gibraltar, one in the European coast and one in the African coast. In both stations the second and third barotropic modes of the Mediterranean basin are well visible (M2, M3). Their energy is much lower than that of the Adriatic modes (about 1,000 times) and, probably for this reason, they are corrected less by the DA. Both stations and many others in the western Mediterranean basin show a third, more energetic peak, which could be a fifth barotropic mode (M5). We can exclude that this peak is a spurious signal from a partial subtraction of the astronomical

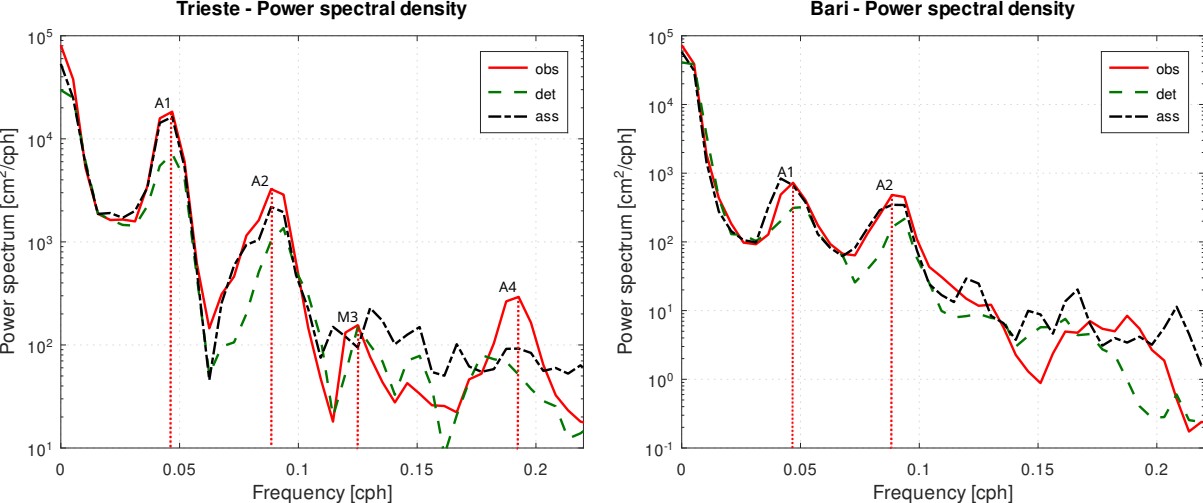

**Figure 14.** Power spectral density of the sea level in Trieste and Bari, in the Adriatic Sea.

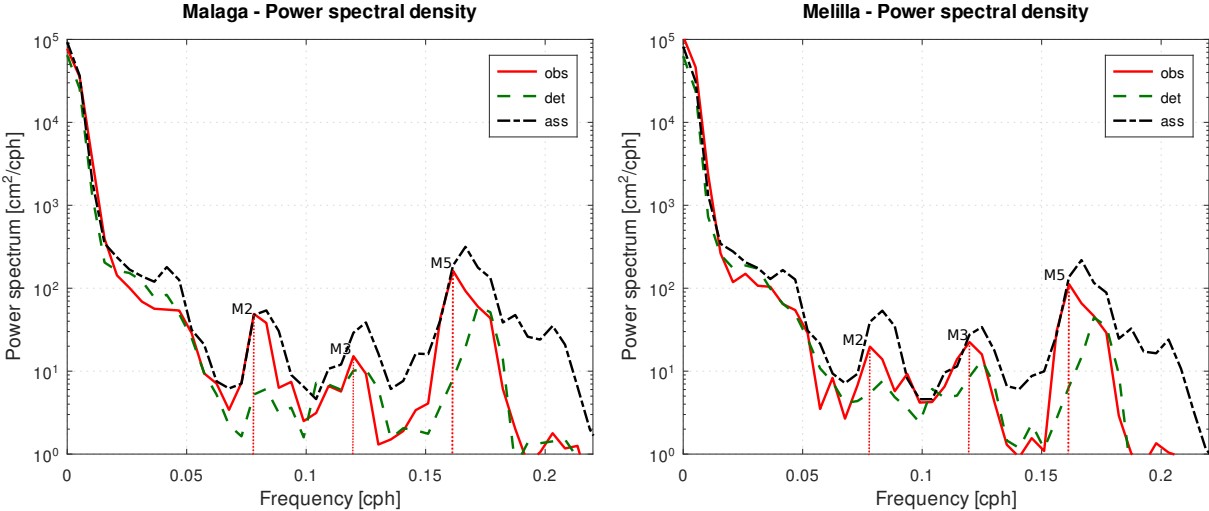

**Figure 15.** Power spectral density of the sea level in Malaga and Melilla, in the western Mediterranean Sea.

tide from the NTR, as it is also present in the surge signal of the model without DA ($SF$). This peak is corrected by the DA even though it is broadened in frequency.

## 4  Discussion


Looking at the results just presented, we can state that DA has an overall positive impact on the reproduction of barotropic sea level signals in the Mediterranean Sea. In the case of the astronomical tide, more than for the other components, the DA has

shown that it can provide an excellent correction of the simulated sea level even in areas very far from the assimilated stations. This fact has been confirmed both in the areas with few stations, such as the eastern Mediterranean and in the open sea. In fact,

although the assimilated stations are all coastal, the altimetric data allowed validation of the tidal results in the open sea. The effectiveness of DA is due to the good number of ensemble members and the good quality of the perturbations. Probably, using localisation techniques the improvement would be weaker, since these techniques limit the correction to areas much closer to the assimilated stations. Without localisation, the analysis increments extends to the entire computational grid (Fig. 13). Furthermore, from a physical point of view, the astronomical tide, as well as the other barotropic components, have large

characteristic spatial lengths which translate into sea level correlations at large distances and in greater spatial effectiveness of the DA. What makes astronomical tide different from surge and seiches is its periodicity and being referred to a mean sea level perfectly constant in time. This avoids any bias in the departures of the assimilation, which are more difficult to deal with the surge and total sea level. These two facts probably contribute to making the astronomical tide results better than those for the other sea level components.

On the contrary, the surge component is not periodic at all and its error mainly depends on the errors in the atmospheric forcings. In the case of the Mediterranean Sea, which is surrounded by a complex orography and often subject to complex meteorological situations, the atmospheric models can have large errors, due to the lack of resolution, of not resolved processes (hydrostatic models) and the lack of high-resolution DA. Their errors result in errors in the surge component which cannot be corrected by the DA in the ocean model when making forecast. However, in the reanalysis simulation, if the assimilation step

is short enough (e.g., hourly), the dynamical system is strongly driven by DA and the error coming from the forcing cannot grow too much.

In the forecast simulations, the DA impact is due to the reduction of the error of the initial state. The error of the initial state propagates over time and sums up the atmospheric forcing error. Analysing the results statistically, the simulations without DA do not show much deterioration from the first to the third forecast day. However, this is not true in the case of extreme

events, when meteorological forcing generally has a larger error. In these cases, even the error of the initial state is often larger, due to pre-existing seiches deriving from previous storms. This error can be corrected by DA and the improvement extends several days, depending on the energy and the damping time of the seiche oscillations. The DA improves not only the error in the seiche part but also those of other sea level components, such as the tidal part (in the total sea level forecast) or the error of surge phenomena which are growing at the time of the analysis. However, in order to catch the formation of a surge in an

operational context, the EnKF should be executed with hourly updates. With one or two updates per day, DA is still a valuable tool to correct the seiche and the tidal parts.

Regarding the computational cost, although there is a need to use a significant number of ensemble members, this is rather low. The ensemble member simulations are perfectly parallel and can run independently between each analysis step. Moreover, barotropic simulations are fast as the equations are quite simple and there is no need to simulate the advection/diffusion of

temperature and salinity. Our workstation is a single-blade mid-level server, with 96 cores and the 81 ensemble members run in parallel. The generation of the ensemble forcings and boundary conditions takes about five minutes, after which the ensemble simulations run parallel except in the 24 analysis steps, where the code is parallelised as well. The total time for carrying out

the entire assimilation procedure is approximately 25 minutes, to which approximately 5 minutes are necessary to carry out five days of forecasting.

Finally, we dedicated the last part to the study of the seiches. In the forecast, we have seen that the DA can lead to a significant improvement, especially where these oscillations are very energetic, as in the Adriatic Sea. As previously mentioned, while in the Adriatic Sea their characteristics are more studied, with the exception of the oscillation A4, which has an unclear origin, they have not been analysed much in the Mediterranean Sea. The observations in our possession confirm and partially correct the periods found in Schwab and Rao (1983), as far as the M2 and M3 modes are concerned. However, we did not detect

the period of the main mode of the Mediterranean Sea, probably because it has not been triggered in the two months that we have analysed, but further investigation is needed. Then, we detected a Mediterranean barotropic oscillation with a period of 6.2 hours, which we called M5, but it is not present in the literature even if its peak is evident in many validation (shown) and calibration (not shown) stations, along the coasts of the western Mediterranean basin. This oscillation, which is more energetic than M2 and M3, is underestimated by the model without DA and, even with the use of DA, it is not reproduced correctly.

Considering that oscillations with a longer period are reproduced better even if less energetic, it is possible that the DA has more difficulty in correcting the high-frequency oscillations. This may be due to the timestep between the observations, every hour, which may be too long to resolve these modes.

## 5   Conclusions

In this paper, we investigated the impact of DA in reproducing the barotropic components of the sea level in the Mediterranean

Sea. We analysed the performances of the model without and with DA in hindcast/reanalysis simulations and in forecast simulations starting from an initial hindcast or analysis state. The barotropic components of the sea level that we considered are the astronomical tide, the surge, with the associated seiches, and the total sea level given by their sum. The results show very good performances of the DA in reanalysis, with the error in the tide reproduction reduced by a third, on average, and slightly worse performances, but always more than good, for the surge and the total sea level. In the case of the surge and the

total sea level, the DA corrects them even in the presence of large errors in the atmospheric forcing, thanks to a sufficiently high assimilation frequency (one hour), a good number of ensemble members and a sufficient number of assimilated observations. The improvements made by the DA in the forecast depend on the reduction of the error of the initial state, but the error coming from the atmospheric forcing (and lateral boundary conditions) cannot be reduced. However, the DA still has a positive impact, especially on the first day forecast, gradually decreasing the following days until the simulations' performances without

DA were reached. The improvement can last longer when seiche oscillations are present. In this case, the initial correction propagates in the following days with a period and decay time equal to those of the triggered barotropic mode. Finally, still considering the forecast simulations, the bias error is lower in the total sea level simulations than in the surge ones.

    In the last part of the results, we have analysed the periods of the barotropic modes of the Adriatic and Mediterranean basins, obtained by the observation power spectra and reproduced by the model. In Adriatic, we detected the periods of the

two main modes (A1, A2), a fourth mode not well known (A4) and the third Mediterranean mode (M3). In the Mediterranean

basin, outside the Adriatic, we detected the periods of the second and third modes (M2, M3) and of a mode that we called M5 (6.2 hours). We tested the reproduction of the associated power peaks by the model in the first-day forecast. While the periods are well reproduced also without DA, the energy of the spectral peaks improves with DA, thus confirming the better reproduction of these oscillations. We noticed also that the DA improves more the low-frequency modes, while it has some difficulties with high-frequency modes. This is probably due to the sampling rate of one hour, which is not enough high.

This work provides a preliminary test of the use of the DA for the reanalysis of tides and surges and seiches in the Mediterranean Sea. Reanalysis simulations can be extended to several years for climatological studies, obtaining good performances as the DA is able to overtake deficiencies in the atmospheric forcing and boundary conditions. Further improvements for the reanalysis, where the error must be low during the whole simulation period, can be obtained using an ensemble Kalman smoother (EnKS). The EnKS is easily applicable to simulation with the EnKF if localisation techniques are not used. Always regarding DA methodologies, an improvement for the reanalysis, but also for the forecast, would be the use of parameter estimation techniques, using an "augmented state" in the EnKF (Evensen, 2009b). The parameter estimation allows to calibrate some model parameters, typically the drag coefficient at the bottom. This method could reduce the model error but then also the DA in its traditional should be used in order to reduce the error of the initial state. Finally, increasing the number of the assimilated observations, from in-situ stations and altimeters, would lead to a further improvement, especially if these are available in areas not well covered. However, while the use of in-situ data is quite easy, as discussed before the altimetric data are difficult to use in storm surge applications (Bajo et al., 2017) and further investigations are needed.

For what concerns the barotropic modes in the Mediterranean and Adriatic basins, some of them are not well understood and their shapes, periods and decay times should be determined with more precision. In this context, DA can be used to provide a reliable reanalysis of the surge from which to extract the seiche part.

The model and the DA system tested in this work will be used, with a similar configuration, in an operational system for forecasting the sea level in several locations of the Mediterranean coasts, with a focus on the Italian coasts. The system will be installed at the ISPRA Centre and will retrieve the observations from the stations along the Italian coast, providing a five-day forecast.

*Code availability.* The hydrodynamic model can be downloaded at: https://github.com/SHYFEM-model/shyfem. The modified version of the model, with the data assimilation code at: https://github.com/marcobj/shyfem

## Appendix A: In-situ coastal stations

In this appendix we report the table with the in-situ stations, their identification numbers and their positions. We used these stations in the paper for the data assimilation and as validation stations.

**Table A1.** List of stations with sea level measurements. The stations with an asterisk are those used in the validation, while the others have been assimilated. The numbering is the one used in the paper and their geographical coordinates are reported as well.

| ID | Lon | Lat | Station | | ID | Lon | Lat | Station |
|---|---|---|---|---|---|---|---|---|
| 1 | -2.930 | 35.290 | Melilla | | 35 | 14.750 | 40.676 | Salerno |
| 2 | -4.417 | 36.711 | Malaga | | 36 | 15.275 | 40.029 | Palinuro |
| 3 | -3.520 | 36.720 | Motril | | 37 | 15.190 | 38.785 | Ginostra |
| 4 | -2.478 | 36.830 | Almeria | | 38 | 8.403 | 40.842 | Porto-Torres |
| 5 | -1.899 | 36.974 | Carboneras | | 39 | 9.114 | 39.210 | Cagliari |
| 6* | -0.973 | 37.596 | Murcia | | 40* | 8.309 | 39.147 | Carloforte |
| 7 | -0.481 | 38.338 | Alicante | | 41* | 13.371 | 38.121 | Palermo |
| 8 | -0.310 | 39.440 | Valencia | | 42 | 13.076 | 37.504 | Sciacca |
| 9* | 1.419 | 38.734 | Formentera | | 43 | 13.526 | 37.285 | Porto-Empedocle |
| 10 | 1.450 | 38.917 | Ibiza | | 44 | 15.093 | 37.498 | Catania |
| 11 | 3.117 | 39.867 | Alcudia | | 45 | 12.604 | 35.499 | Lampedusa |
| 12 | 1.213 | 41.078 | Tarragona | | 46 | 17.137 | 39.083 | Crotone |
| 13 | 2.160 | 41.340 | Barcelona | | 47 | 17.223 | 40.475 | Taranto |
| 14 | 3.107 | 42.520 | Port-Vendres | | 48 | 18.497 | 40.147 | Otranto |
| 15 | 3.699 | 43.397 | Sete | | 49 | 16.866 | 41.140 | Bari |
| 16 | 4.893 | 43.405 | Fos-sur-Mer | | 50* | 16.177 | 41.888 | Vieste |
| 17 | 5.914 | 43.122 | Toulon | | 51 | 15.501 | 42.119 | Tremiti |
| 18 | 6.717 | 43.359 | Port-Ferreol | | 52 | 14.414 | 42.355 | Ortona |
| 19* | 6.933 | 43.483 | La-Figueirette | | 53 | 13.890 | 42.960 | San-Benedetto-del-Tronto |
| 20 | 7.421 | 43.728 | Monaco | | 54 | 13.506 | 43.624 | Ancona |
| 21 | 9.350 | 42.967 | Centuri | | 55 | 12.282 | 44.492 | Ravenna |
| 22 | 8.938 | 42.635 | Ile-Rousse | | 56 | 12.426 | 45.418 | Venice |
| 23 | 8.760 | 41.920 | Ajaccio | | 57* | 12.511 | 45.313 | AAOT |
| 24* | 9.374 | 41.836 | Solenzara | | 58 | 13.757 | 45.649 | Trieste |
| 25 | 8.018 | 43.878 | Imperia | | 59 | 21.319 | 37.640 | Katakolo |
| 26 | 8.870 | 44.380 | Genova | | 60* | 23.621 | 37.935 | Peiraias |
| 27* | 9.857 | 44.096 | La-Spezia | | 61 | 24.941 | 37.438 | Syros |
| 28 | 10.299 | 43.546 | Livorno | | 62* | 35.653 | 34.242 | Batroun |
| 29 | 10.238 | 42.742 | Marina-di-Campo | | 63* | 29.879 | 31.209 | Alexandria |
| 30 | 11.789 | 42.093 | Civitavecchia | | | | | |
| 31 | 12.634 | 41.446 | Anzio | | | | | |
| 32 | 12.965 | 40.895 | Ponza | | | | | |
| 33* | 13.589 | 41.209 | Gaeta | | | | | |
| 34 | 14.269 | 40.841 | Napoli | | | | | |

*Author contributions.* Marco Bajo: writing and reviewing the paper, developing the data assimilation code and its binding to SHYFEM, running some simulations, processing the results. Christian Ferrarin: Reviewing the paper, running some simulations, processing the results. Georg Umgiesser: Developing the SHYFEM code. Andrea Bonometto and Elisa Coraci: Providing the atmospheric forcing, financial support.

*Competing interests.* We do not have any competing interests.

*Acknowledgements.* We thank ISPRA for supporting the development of an operational system for forecasting the sea level along the Italian coasts.

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
