# Peer review of "Modelling the barotropic sea level in the Mediterranean Sea using data assimilation"

_EGUsphere, 2022_

## Author Comment (AC1)

*Reviewer 1*

*The authors investigated the potential role of Data Assimilation in improving the accuracy of barotropic processes induced variant scale/mode sea level anomaly in the Mediterranean Sea. The study is based on the state-of-the-art simulation kernel in SHYFEM. The authors comprehensively investigated the improvement of the astronomical tide, surge and seiches implemented by DA, and promoted the adaptability of SHYFEM with inclusion of EnKF. The manuscript is well written and organized with a sensible logic. However, given I still have these several following major concerns, I cannot recommend an acceptance at its present form.*

We thank the Reviewer for the helpful comments, which will improve the quality of the paper. We answer the individual points below.

- *Although it is still a nowadays great challenge to DA to treat/improve the hindcast and forecast of sea level anomaly in the region where the SLA oscillation is significant, I'm still wondering why the authors conduct this simulation in a two-dimensional or barotropic configuration? Will the inclusion of, e.g. dynamic height associated with the baroclinic processes be really negligible in the region? If it is not, why the heat fluxes, evaporation and precipitation, as well as riverine discharges are excluded? The larger scale circulation, at least those in the synoptic scale, is another issue related to this concern. Could the authors include some discussion related to the unimportance of these processes? Or, the authors may want to state that they are treating those larger-scaled motions as reference levels already, although I don't think that is a straightforward statement.*

  This comment has points in common with the second Reviewer's fourth comment, so please read that explanation as well.

  Actually, this point is not sufficiently clarified in the paper; therefore, a detailed explanation will be added in the Introduction. Sea level is one of the most complex oceanographic variables to model, due to the innumerable components it can have. For this reason, we explicitly wrote in the title that we analyse the "barotropic" sea level, not the baroclinic component, thus using the model in the barotropic version. With the term "barotropic sea level" we refer to the barotropic tide, the storm surge and the total level made up of these two components (and the oscillations of the barotropic modes - seiches - triggered by the storm surge). Indeed, the storm-surge definition is sometimes not unique. In this study and following Pugh (1996), we consider the storm surge as the sea level induced by the effect of the wind stress and the Inverse Barometric Effect

  There are, however, several other factors influencing the sea. The contribution of the river run-off may not be negligible in some specific coastal areas, such as deltas and estuaries (e.g., Mississippi river, Bangladesh coast). However, in the Mediterranean Sea, the conditions are very far from those present in these places.

  As for the baroclinic processes related to vertical and horizontal temperature and salinity gradients, forced by (sensible, latent) heat fluxes, rain, freshwater inputs from rivers, and the associated SLA, they are not investigated in this paper, as stated in the title. However, these processes follow much larger timescales of variation and, to our knowledge, are never considered in storm surge or tidal models in European seas (see the papers cited below).

The effectiveness of barotropic models to reproduce storm surges and tides is also proved by the corrections usually applied to SLA altimeter data. The storm surge part is removed using the (2D barotropic) Mog2D model with this motivation: "The high-frequency oceanic signal (pressure and wind meteorological forcing) is badly sampled by altimeter measurements" - https://www.aviso.altimetry.fr/en/data/products/auxiliary-products/dynamic-atmospheric-correction/description-atmospheric-corrections.html). The astronomical tide is instead subtracted through another barotropic 2D model, the FES2014.

As regards the effects of the synoptic circulation on the sea level, the synoptic ocean signal is filtered into the Mediterranean through the Strait of Gibraltar. It consists of low-frequency oscillations (Bajo et al., 2019) and it is included in the boundary conditions. As regards the synoptic atmospheric circulation, this is present in the wind and pressure forcings. We will add in the paper a further explanation of the boundary conditions.

The above-mentioned arguments justify the use of the 2D approach in simulating and forecasting storm surges and tides. In the Adriatic Sea, the most extreme events that occurred in 1966, 2018 and 2019, were successfully modelled using SHYFEM in a configuration analogous to the one used in this paper (Roland et al., 2009; Cavaleri et al., 2019; Ferrarin et al., 2021). Furthermore, the model, always in barotropic configuration, has been used for over ten years at the centre for forecasting and warning of high tides in Venice (Bajo et al., 2007). SHYFEM in 2D barotropic version has been effectively used for the reproduction of the seiche oscillations (Bajo et al., 2019) and for the study of the astronomical tide (Ferrarin et al., 2018). A similar 2D barotropic approach was successfully used by Fernández-Montblanc et al. (2019) with the SCHISM model to reproduce storm surge, tide and the total level (barotropic) in several European seas. In Xavier et al., (2014) SELFE in 2D barotropic version is used to reproduce the storm Xynthia, one of the most extreme ever recorded in Europe. Similar models are also used elsewhere, still in Europe, by several storm surge and tide researchers (e.g., Flowerdew et al., 2010, Horsburgh et al., 2021). Regarding the astronomical tide, as mentioned before the various versions of FES use a 2D barotropic model.

To conclude, in the next version of the paper, the focus of the article will be better explained in the introduction (the barotropic component, as written in the title) and a part will be added in the introduction to better analyse the various contributions of the sea level and adding the citations presented here.

- *I still have concerns about how did the simulation treat the open boundary condition, although the manuscript did clarify that the authors treated the boundary condition with great effort. If sea level is kind of prescribed at the western boundary, how could the circulation (including their impacts in SLA and currents) be connected with that to the further west of the open boundary, which I think is provided by, for example, the CMEMS reanalyses. I may also suggest the authors include a paragraph to elaborate the way the open boundary condition is implemented or explicitly show the algorithm of the open boundary condition.*

The boundary conditions are described in section 2.1. Actually, they are described shortly. As suggested we will extend the description, evaluating

whether to introduce a subsection for boundary conditions and, perhaps, for surface forcings. In the paper we made two mistakes in describing the boundary condition:

- we said we applied the conditions to Gibraltar, but the model grid ends at -7.2W, in the Atlantic Ocean. This allows for a fairly distant boundary from the Mediterranean Sea.

- The link specified in the paper refers to the reanalysis of the CMEMS model, while we have used the analysis/forecast product (https://doi.org/10.25423/CMCC/MEDSEA_ANALYSISFORECAST_PHY_006_013_EAS7).

- *Why the satellite altimetry data is not used as observed data in this research? Are they at least usable for the astronomical tide correction and forecast? If gridded data is problematic, how about the along-track data? There are dataset of harmonic constants extracted from the along-track data by using this operation, and the authors mainly used much higher resolution records at the surrounding tidal gauge. I mean, there are more observations with much higher spatial coverage may help further improved the DA.*

This comment has similarities with the second Reviewer's fifth comment, so please read that explanation as well. This is a good suggestion, we thank the Reviewer for pointing it out to us. Indeed we can validate the model not only in the validation stations but also where the altimeter harmonic constants are available. We will therefore add a validation part of the astronomical tide based on altimetric data if these will have good quality in the Mediterranean Sea.

- *In the perturbation runs, why the drag coefficient Cd in the quadratic formulation is not perturbed? Dissipation of energy with the scales smaller than tides through the bottom friction could also be an important process that determines the characteristics of tidal currents, and in this sense, although the authors stated that the current research is focusing on SLA variations, in the current configuration, accuracy in flows will also be an important aspect.*

We thank the Reviewer for noting this. Actually, the Cd was perturbed, but we forgot to write it. In addition to the Cd, we also perturbed a calibration factor for the calculation of the loading tide (called ltidec in SHYFEM) in the simulations using the tidal potential (tide and total level). For both parameters, in each simulation, the 80 perturbations belonging to a Gaussian distribution are calculated, centred at 0.0025 (Cd) and 6.e-05 (ltidec), with a standard deviation of 0.0005 (Cd) and 1.e-05 (ltidec). As commented by the Reviewer, Cd has great importance in the dissipation of energy and therefore also in the correct reproduction of the levels. Also, since tide-only simulations do not have atmospheric forcing, in this case, Cd and ltidec are even more important to create an ensemble with a wide enough spread. We will add new paragraphs to section 2.3 describing what is reported here.

*Did the authors analyze whether the current design could also improve flows or not?*

We noted a change in water transports compared to the simulation without DA but we did not compare them to any measures. However, if the cross-correlation between levels and currents is correct (the size of the ensemble - 81 members - should be sufficient), then currents should improve as well. On a smaller scale,

we had seen improvements in the current by assimilating sea-level data in the inlets of the Lagoon of Venice (Ferrarin et al., 2021).

- *In my opinion, it is still important to rely on DA to improve the parameterization in the simulation, since it is not that feasible for operational users to generate a large number of perturbation runs to have that short-term forecast improved.*

The parameter estimation technique with the DA is very interesting and we had taken it into consideration. Although we haven't currently developed the necessary code, it shouldn't be very complicated to do in the future, and we could use it to estimate Cd or the loading-tide coefficient with spatial variability. We will discuss this future development in the conclusions.

Regarding the operational use of an ensemble, there are already several examples of operational systems with ensemble DA much more computationally heavy than the system presented here. For example, the CMEMS model for the Arctic (Topaz, https://doi.org/10.48670/moi-00001) uses 100 members in a 3D baroclinic model. Also, Ohishi et al. (2022) use 100 members. We are currently implementing the system described in this study for operational use at the Italian Institute for Environmental Protection and Research, ISPRA. The computational server has 96 cores, so that the 80+1 ensemble simulations run perfectly in parallel (the code allows this), with no slowdown compared to a single simulation. In a daily simulation, 24 analysis steps are performed, each taking about 30 seconds. Finally, it takes about 5 minutes in the beginning to create the perturbed atmospheric forcing and the perturbed boundary condition. In total, the 81 2D barotropic simulations with DA take about 25 minutes to provide a final analysis state, which is more than reasonable for operational purposes. Such a fast modelling system can be applied several times per day using real-time observations for improving the forecasts (ideally each time new observations are available).

Although the parameter estimation technique can bring improvements in the model error, it is also necessary to use DA in the traditional way, to improve the initial state, especially in the case of seiche oscillations, as discussed in this paper.

- *It is really hard to intensify the meshes in Figure 1. Could you zoom in to some critically locations to show the spatial variability of resolution?*

Indeed, it is difficult to distinguish the resolution of the grid. As suggested we will zoom the grid in some areas (probably north Adriatic, Gibraltar).

Bibliography

M. Bajo, L. Zampato, G. Umgiesser, A. Cucco, P. Canestrelli, A finite element operational model for storm surge prediction in Venice, Estuarine, Coastal and Shelf Science, Volume 75, Issues 1–2, 2007, Pages 236-249, https://doi.org/10.1016/j.ecss.2007.02.025.

Marco Bajo, Francesco De Biasio, Georg Umgiesser, Stefano Vignudelli, Stefano Zecchetto, Impact of using scatterometer and altimeter data on storm surge forecasting, Ocean Modelling, Volume 113, 2017, Pages 85-94, https://doi.org/10.1016/j.ocemod.2017.03.014.

Bajo, M, Međugorac, I, Umgiesser, G, Orlić, M. Storm surge and seiche modelling in the Adriatic Sea and the impact of data assimilation. Q J R Meteorol Soc. 2019; 145: 2070–2084. https://doi.org/10.1002/qj.3544

L. Cavaleri, M. Bajo, F. Barbariol, M. Bastianini, A. Benetazzo, L. Bertotti, J. Chiggiato, S. Davolio, C. Ferrarin, L. Magnusson, A. Papa, P. Pezzutto, A. Pomaro, G. Umgiesser, The October 29, 2018 storm in Northern Italy – An exceptional event and its modeling, Progress in Oceanography, Volume 178, 2019, https://doi.org/10.1016/j.pocean.2019.102178.

T. Fernández-Montblanc, M.I. Vousdoukas, P. Ciavola, E. Voukouvalas, L Mentaschi, G. Breyiannis, L. Feyen, P. Salamon, Towards robust pan-European storm surge forecasting, Ocean Modelling, Volume 133, 2019, Pages 129-144, https://doi.org/10.1016/j.ocemod.2018.12.001.

Christian Ferrarin, Debora Bellafiore, Gianmaria Sannino, Marco Bajo, Georg Umgiesser, Tidal dynamics in the inter-connected Mediterranean, Marmara, Black and Azov seas, Progress in Oceanography, Volume 161, 2018, Pages 102-115, https://doi.org/10.1016/j.pocean.2018.02.006.

Ferrarin, C., Bajo, M., and Umgiesser, G.: Model-driven optimization of coastal sea observatories through data assimilation in a finite element hydrodynamic model (SHYFEM v. 7_5_65), Geosci. Model Dev., 14, 645–659, https://doi.org/10.5194/gmd-14-645-2021, 2021.

Christian Ferrarin, Marco Bajo, Alvise Benetazzo, Luigi Cavaleri, Jacopo Chiggiato, Silvio Davison, Silvio Davolio, Piero Lionello, Mirko Orlić, Georg Umgiesser, Local and large-scale controls of the exceptional Venice floods of November 2019, Progress in Oceanography, Volume 197, 2021, https://doi.org/10.1016/j.pocean.2021.102628.

Flowerdew, J., Horsburgh, K., Wilson, C. and Mylne, K. (2010), Development and evaluation of an ensemble forecasting system for coastal storm surges. Q.J.R. Meteorol. Soc., 136: 1444-1456. https://doi.org/10.1002/qj.648

Horsburgh, K., Haigh, I.D., Williams, J. et al. "Grey swan" storm surges pose a greater coastal flood hazard than climate change. Ocean Dynamics 71, 715–730 (2021). https://doi.org/10.1007/s10236-021-01453-0

Ohishi, S., Hihara, T., Aiki, H., Ishizaka, J., Miyazawa, Y., Kachi, M., and Miyoshi, T.: An ensemble Kalman filter system with the Stony Brook Parallel Ocean Model v1.0, Geosci. Model Dev., 15, 8395–8410, https://doi.org/10.5194/gmd-15-8395-2022, 2022

Pugh, D.T. (1996) Tides, surges and mean sea-level (reprinted with corrections) , Chichester, UK. John Wiley & Sons, Ltd., 486pp.

Aron Roland, Andrea Cucco, Christian Ferrarin, Tai-Wen Hsu, Jian-Ming Liau, Shan-Hwei Ou, Georg Umgiesser, Ulrich Zanke, On the development and verification of a 2-D coupled wave-current model on unstructured meshes, Journal of Marine Systems, Volume 78, Supplement, 2009, Pages S244-S254, https://doi.org/10.1016/j.jmarsys.2009.01.026.

Xavier Bertin, Kai Li, Aron Roland, Yinglong J. Zhang, Jean François Breilh, Eric Chaumillon, A modeling-based analysis of the flooding associated with Xynthia, central

Bay of Biscay, Coastal Engineering, Volume 94, 2014, Pages 80-89, https://doi.org/10.1016/j.coastaleng.2014.08.013.

---

## Author Comment (AC2)

*Reviewer 2*

*The manuscript presents the predictive capability of a 2D barotropic model of the Mediterranean Sea sea level with and without the assimilation of the observations obtained from coastal tide gauges stations. The hydrodynamical model setup and ensemble Kalman filter based data assimilation system is described along with the perturbation schemes applied for ensemble generation. The results are presented for the total sea level as well as different contributions from the astronomical tides, surge and seiche for the hindcast/analysis and forecast periods for the December 2019 seiche occurrence following the November 2019 extreme event in the Adriatic Sea.*

*The manuscript requires a substantial revision before publication. Below are major comments and minor suggestions.*

We thank the Reviewer for the dedicated time and detailed review, which helps to increase the quality of this work. Below we provide the answer to the individual points.

*Major comments*

*To start with, for the readability of the manuscript, I suggest including a table of experiments to make it easier to follow, especially the results section. A flow chart for the production cycle would also help since it is difficult to understand where the hindcast/analysis ends and where the forecast starts. This may also help for future works since this system is proposed as a candidate for operational forecasting.*

As suggested, we will add a table summarising all the numerical experiments carried out, before showing the results. We will also add a flowchart of the DA procedure and simulations, in the analysis and forecast.

*Moreover, the terminology used can be improved. There are terms used interchangeably such as analysis, reanalysis, hindcast simulation with data assimilation. I suggest homogenising them for an easier read and paying attention throughout the text to use the terminology that is already established, such as using analysis ensemble mean instead of average analysis state.*

Indeed, there is some confusion in the terminology, especially when referring to simulations in the hindcast period. We will pay more attention to the terminology used, defining it correctly.

*Secondly, I understand that the manuscript targets seiche in December 2019 however, it would be nice to see the evolution of the error in the sea level over a longer period given that the current version of the model is quite cheap as stated by the authors. I expected at least to see some analysis and the skill of the model in the November 2019 high tide event in the northern Adriatic Sea which resulted in the flooding of the city of Venice.*

Indeed, we showed December's event since DA has an effect mainly on seiches. However, as noted by the Reviewer, it can be interesting also to look at the behaviour in November's storm-surge event. In that event, the 2-day forecast wind totally missed the event, while the 1-day wind was good. DA cannot correct of course the 2-day forecast, while the 1-day forecast did not need a correction. However, the high tide of the following days (due to seiches) was better reproduced by DA. As suggested by the Reviewer, we will add this event to the results.

*On the other hand, SHYFEM is shown to be a skillful model in various previous studies. It is hard to understand why a simplified version is used in a development that is a candidate for an operational forecasting system. I think that in the cases where the errors and bias are large there is missing the steric steric part from the thermohaline contribution to sea level variability. This should be clarified and justified.*

We thank the Reviewer for this comment which is similar to the first Reviewer's first comment. Please refer back also to that answer, which has similar points. The version of SHYFEM we used is one of the latest available, and there is no simplification on the numeric or other parts of the code. However, the baroclinic (steric) part was not activated. This is usual in storm-surge and tide modelling since to reproduce the barotropic sea-level components a barotropic model should be used (see the references in the 1st common of 1st rev). As we wrote in the title, this work focuses on the barotropic components (surges, tides, seiches) and not on the baroclinic part of the sea level.

Steric components are usually not considered in storm surge models as their variation is much slower and almost constant in 5-day forecasts (see references and comments on 1st point, 1st rev.).

Finally, we should consider the execution speed and the complexity of the model and of the DA procedure. Usually, 2D barotropic models are over 10 times faster than baroclinic ones. For example, in a simulation with SHYFEM in the Venetian lagoon, the model takes 16 minutes in a 2D barotropic simulation and 176 minutes in a baroclinic simulation with 12 vertical levels (however the Mediterranean would need at least 30 vertical levels). The simulated water levels from the two simulations are almost indistinguishable. Regarding the DA part, a baroclinic model would also require the perturbation of all the other forcings (solar radiation, air temperature, relative humidity, cloudiness, rain) and boundary conditions (T, S), which would take time. Assimilating only sea level would be not advisable, and  T, S data should be assimilated as well.

In the revised manuscript we will add a similar explanation in the introduction, as well as some bibliography (see 1st point, 1st rev).

*Finally, it is not easy to completely grasp the improvements brought by the data assimilation of observations from tide gauges since they are limited in space coverage. Satellite observations could be used at least for validation to see the impact, if not assimilated. The results should be supported by maps of, for example, mean dynamic topography, increments. I think there may be other resources for the coastal sea level data for assimilation such as Copernicus Marine, SeaDataNet or EMODNet to better cover the eastern basin.*

This comment has points in common with the third comment by the first Reviewer, so please read that explanation as well.

Altimeter data could actually be useful in areas with few coastal stations, such as the eastern basin of the Mediterranean Sea. However, using altimetry data to correct high-frequency sea-level signals is problematic, as pointed out also in: https://www.aviso.altimetry.fr/en/data/products/auxiliary-products/dynamic-atmospheric-correction/description-atmospheric-corrections.html. And this is the reason why the storm surge signal is removed from SLA products and is rarely assimilated. Note also

that in order to remove the storm surge and tidal signals, 2D barotropic models are used (Mog2D, FES2014). In the past, we made some attempts to assimilate altimetric data (reintroducing the Mog2D correction) in a project with ESA on storm surges (see: Bajo et al. 2017). The results were modest and slightly positive only in the case of "lucky" tracks (direction along the Adriatic Sea and right time). Anyway, this is rare in the Mediterranean Sea, as the ratio between coastal areas, where the altimetric signal is scarce, and the open sea is very high. There are also issues related to the reference sea level, SLA refers to the Mean Sea Surface, different from model msl and in-situ data msl.

As future progress, we will try again to assimilate altimetric data, considering that they are improving in coastal areas. The imminent launch of the SWOT satellite (https://www.aviso.altimetry.fr/en/missions/future-missions/swot.html) cloud also be of interest, as its swat is large (about 120km). We will add some of this in the paper.

As suggested also by the first Reviewer, the use of altimetric data to determine the tidal harmonic constants is interesting. We will try to use them to validate the mode data (if the quality is good enough in the Mediterranean Sea).

*Minor suggestions*

*Title:  Mediterranean -> Mediterranean Sea*

Ok.

*L27  "easily predictable" -> please refer to the sources of uncertainty in the estimates of tides e.g. bathymetry*

Ok.

*L92 Please be more precise about the mesh resolution and give a measure of change from the open ocean to the coastal seas. Danilov (2022) may help. https://agupubs.onlinelibrary.wiley.com/doi/full/10.1029/2022MS003177*

Ok.

*L101 as done with the atmospheric forcing product, please cite the Copernicus Marine multi-year product explicitly in the references, not only with DOI. It should be clarified why the authors used the multi-year product for the lateral open boundary conditions in the Atlantic Ocean while a NRT analysis/forecast product as in the atmospheric forcing is available in Copernicus Marine catalogs with tides for the experiment period. This is also one of the parameters that defines the type of experiment performed: an analysis, a reanalysis etc…*

We thank the Reviewer for noting this. Actually, the DOI in the paper is wrong, we used the analysis/forecast product (https://doi.org/10.25423/CMCC/MEDSEA_ANALYSISFORECAST_PHY_006_013_EAS 7). The reanalysis product does not even cover the lateral edge of our grid (-7.2W). Furthermore, as also reported in the comments to the first Reviewer, in the paper we wrote that the boundary conditions were imposed in Gibraltar, while the grid ends in the Atlantic Ocean. In the next version, we will correct these errors and expand the description of the boundary conditions. We will cite better the CMEMS product.

*L102 please explain how you de-tide the sea level.*

The sea level without the astronomical component (de-tided) is supplied directly by CMEMS. We will explain this point better in the paper.

*L124 missing citation in the parentheses. Please add it.*

Ok.

*L128 Please add the mean sea level map and compare with the MDT products such as MDT-CMEMS_2020_MED in Copernicus Marine Catalog*

In the paper we referred to the MDT of the model, however, this is incorrect since the model simulates only the barotropic part of the sea level. We would therefore like to change the nomenclature to the mean sea level of the model.

*L145 please justify 2 cm of observational error, is it only the instrumental error considered? How do the increments with such a small observational error look like? A map of increments may help to see whether there is an overfitting.*

The stations' sensors have an accuracy much lower than 1cm (radar sensors, see e.g.: https://www.mareografico.it/?session=0S1476768288B907168WO8287&syslng=ita&sysmen=-1&sysind=-1&syssub=-1&sysfnt=0&code=SENS&idse=C). Some have pressure sensors with an error of about 1cm. We used a 2cm error precisely to avoid overfitting. The errors are still low, but the results obtained are good and they don't seem to have overfitting problems (we had also done some tests with 1-3cm).

*L153 grid -> node*

Ok.

*L153 "A_a^* is that of the analysis states not corrected". What do you mean? The definition of analysis implies a corrected background. Do you mean background?*

$A\_a^*$ is the original analysis, $A\_a$ is the quantity in the formula 2. The text is incorrect, we will modify it.

*L156 "levels" -> of what?*

The variable sea level ($\zeta$) in equations 1. Will be explained further.

*L162 "average analysis state" -> analysis ensemble mean*

Ok.

*L169 Please justify 400 km. For example, Sakov et al. 2012 chose 250 km in a north Atlantic - Arctic Ocean system using the same methodology.*

As in Sakov et al. (2012), our justification is mainly empirical. We saw that 400km gives good results. Anyway, 400km SLP perturbations produce sub-synoptic systems that are of the same length scale as the real ones in the Mediterranean Sea (see e.g., 22 November 2022, 06 UTC – https://www1.wetter3.de/archiv_dwd_dt.html). We will add a sentence to the text.

*L192 This is the definition of analysis ensemble mean. Please use it.*

Ok.

The concept is badly explained, we will try to write it better. For example, if we wanted the astronomical tide in 2025, we could compute tidal level time series from the harmonic constants in some locations where they are known with accuracy. Later, we could use these observations to run the DA and to make an analysis of the 2025 year.

This sentence is also unclear, we will change it. We meant that we display the first three days of the forecast in the results, not the fourth and fifth, because they would lengthen the section and the advantages of the DA on the fourth and fifth days are reduced.

Ok, we will make a flowchart.

*L222 What are the parameters of DA? Inflation and localization?*

We have always used adaptive inflation. The parameters that we varied in the tests are the localisation with different radii (in the end we did not use localisation), the number of ensemble members, the method (EnKF, EnSRF), the observation error and the C-G radius in the formula 2. We will write better in the text.

*L223 Local analysis is only one way of localization.*

Thank you to note it, we will correct it. We tested Domain Localization and not Covariance Localisation.

*L234 Looks like too big error (9.3 cm) even for a free model and with a 2 cm of observation error reduces to only 3.6 cm. Is it because the barotropic model is missing the steric contribution? Please compare with altimeter products.*

Actually, as the Reviewer noted, such error is partly due to the fact that the barotropic model is missing the steric contribution and partly to the fact that November-December 2019 was a period in which storm surges and seiches were particularly intense.

*L347 "is not present in our observations"? Do you mean in the period of observations used?*

Yes, we will write better this sentence. We didn't see this peak in our spectral analyses. A reason could be that this mode has not been triggered, it should be not very energetic and we didn't know any previous work observing it.

*L352 There are other sources of error in DA besides model and representativeness error. Please correct.*

This sentence does not fit well into the discussion, we will decide whether to remove it or change it also considering the other sources of error.

Bibliography

M. Bajo, L. Zampato, G. Umgiesser, A. Cucco, P. Canestrelli, A finite element operational model for storm surge prediction in Venice, Estuarine, Coastal and Shelf Science, Volume 75, Issues 1–2, 2007, Pages 236-249, https://doi.org/10.1016/j.ecss.2007.02.025.

Marco Bajo, Francesco De Biasio, Georg Umgiesser, Stefano Vignudelli, Stefano Zecchetto, Impact of using scatterometer and altimeter data on storm surge forecasting, Ocean Modelling, Volume 113, 2017, Pages 85-94, https://doi.org/10.1016/j.ocemod.2017.03.014.

Bajo, M, Međugorac, I, Umgiesser, G, Orlić, M. Storm surge and seiche modelling in the Adriatic Sea and the impact of data assimilation. Q J R Meteorol Soc. 2019; 145: 2070–2084. https://doi.org/10.1002/qj.3544

L. Cavaleri, M. Bajo, F. Barbariol, M. Bastianini, A. Benetazzo, L. Bertotti, J. Chiggiato, S. Davolio, C. Ferrarin, L. Magnusson, A. Papa, P. Pezzutto, A. Pomaro, G. Umgiesser, The October 29, 2018 storm in Northern Italy – An exceptional event and its modeling, Progress in Oceanography, Volume 178, 2019, https://doi.org/10.1016/j.pocean.2019.102178.

T. Fernández-Montblanc, M.I. Vousdoukas, P. Ciavola, E. Voukouvalas, L Mentaschi, G. Breyiannis, L. Feyen, P. Salamon, Towards robust pan-European storm surge forecasting, Ocean Modelling, Volume 133, 2019, Pages 129-144, https://doi.org/10.1016/j.ocemod.2018.12.001.

Christian Ferrarin, Debora Bellafiore, Gianmaria Sannino, Marco Bajo, Georg Umgiesser, Tidal dynamics in the inter-connected Mediterranean, Marmara, Black and Azov seas, Progress in Oceanography, Volume 161, 2018, Pages 102-115, https://doi.org/10.1016/j.pocean.2018.02.006.

Ferrarin, C., Bajo, M., and Umgiesser, G.: Model-driven optimization of coastal sea observatories through data assimilation in a finite element hydrodynamic model (SHYFEM v. 7_5_65), Geosci. Model Dev., 14, 645–659, https://doi.org/10.5194/gmd-14-645-2021, 2021.

Christian Ferrarin, Marco Bajo, Alvise Benetazzo, Luigi Cavaleri, Jacopo Chiggiato, Silvio Davison, Silvio Davolio, Piero Lionello, Mirko Orlić, Georg Umgiesser, Local and large-scale controls of the exceptional Venice floods of November 2019, Progress in Oceanography, Volume 197, 2021, https://doi.org/10.1016/j.pocean.2021.102628.

Flowerdew, J., Horsburgh, K., Wilson, C. and Mylne, K. (2010), Development and evaluation of an ensemble forecasting system for coastal storm surges. Q.J.R. Meteorol. Soc., 136: 1444-1456. https://doi.org/10.1002/qj.648

Horsburgh, K., Haigh, I.D., Williams, J. et al. "Grey swan" storm surges pose a greater coastal flood hazard than climate change. Ocean Dynamics 71, 715–730 (2021). https://doi.org/10.1007/s10236-021-01453-0

Ohishi, S., Hihara, T., Aiki, H., Ishizaka, J., Miyazawa, Y., Kachi, M., and Miyoshi, T.: An ensemble Kalman filter system with the Stony Brook Parallel Ocean Model v1.0, Geosci. Model Dev., 15, 8395–8410, https://doi.org/10.5194/gmd-15-8395-2022, 2022

Pugh, D.T. (1996) Tides, surges and mean sea-level (reprinted with corrections) , Chichester, UK. John Wiley & Sons, Ltd., 486pp.

Aron Roland, Andrea Cucco, Christian Ferrarin, Tai-Wen Hsu, Jian-Ming Liau, Shan-Hwei Ou, Georg Umgiesser, Ulrich Zanke, On the development and verification of a 2-D coupled wave-current model on unstructured meshes, Journal of Marine Systems, Volume 78, Supplement, 2009, Pages S244-S254, https://doi.org/10.1016/j.jmarsys.2009.01.026.

Xavier Bertin, Kai Li, Aron Roland, Yinglong J. Zhang, Jean François Breilh, Eric Chaumillon, A modeling-based analysis of the flooding associated with Xynthia, central Bay of Biscay, Coastal Engineering, Volume 94, 2014, Pages 80-89, https://doi.org/10.1016/j.coastaleng.2014.08.013.

---

## Author Response (AR1)

*Reviewer 1*

*The authors investigated the potential role of Data Assimilation in improving the accuracy of barotropic processes induced variant scale/mode sea level anomaly in the Mediterranean Sea. The study is based on the state-of-the-art simulation kernel in SHYFEM. The authors comprehensively investigated the improvement of the astronomical tide, surge and seiches implemented by DA, and promoted the adaptability of SHYFEM with inclusion of EnKF. The manuscript is well written and organized with a sensible logic. However, given I still have these several following major concerns, I cannot recommend an acceptance at its present form.*

We thank the Reviewer for the helpful comments, which will improve the quality of the paper. For the answer, we refer also to the first answer that we provided online.

- *Although it is still a nowadays great challenge to DA to treat/improve the hindcast and forecast of sea level anomaly in the region where the SLA oscillation is significant, I'm still wondering why the authors conduct this simulation in a two-dimensional or barotropic configuration? Will the inclusion of, e.g. dynamic height associated with the baroclinic processes be really negligible in the region? If it is not, why the heat fluxes, evaporation and precipitation, as well as riverine discharges are excluded? The larger scale circulation, at least those in the synoptic scale, is another issue related to this concern. Could the authors include some discussion related to the unimportance of these processes? Or, the authors may want to state that they are treating those larger-scaled motions as reference levels already, although I don't think that is a straightforward statement.*

  - Due to the comments of the reviewer we changed the introduction and the definition of the barotropic variations of the sea level (tide, surges and seiches), which are the focus of this paper. We hope that in this way our methodology becomes clearer (rows 15-22, 58-60). With this model configuration we cannot and we won't reproduce SLA due to the baroclinic part, which however varies with a timescale of weeks.

  - In the new section describing the altimeter data that we used (section 2.2.2) we discuss the difficulty to use altimeter data for storm surge applications (actually we used them in a past paper);

  - In section 2.1, in the description of the model, at the end we cite several works, using a similar model configuration, which proves the suitability of this formulation for tides, surges and seiches reproduction. We provided both papers using the SHYFEM model and papers using other models by many different experienced researchers (rows 112-124).

- *I still have concerns about how did the simulation treat the open boundary condition, although the manuscript did clarify that the authors treated the boundary condition with great effort. If sea level is kind of prescribed at the western boundary, how could the circulation (including their impacts in SLA and currents) be connected with that to the further west of the open boundary, which I think is provided by, for example, the CMEMS reanalyses. I may also suggest the authors include a paragraph to elaborate the way the open boundary condition is implemented or explicitly show the algorithm of the open boundary condition.*

As suggested, we added a section with the description of the forcing and boundary condition (section 2.1.1). Now, in this section, we provide a better description and we have also corrected the reference to the CMEMS model used for the boundary conditions (rows 125-142).

- *Why the satellite altimetry data is not used as observed data in this research? Are they at least usable for the astronomical tide correction and forecast? If gridded data is problematic, how about the along-track data? There are dataset of harmonic constants extracted from the along-track data by using this operation, and the authors mainly used much higher resolution records at the surrounding tidal gauge. I mean, there are more observations with much higher spatial coverage may help further improved the DA.*

We thank the Reviewer for this suggestion. We downloaded the Aviso Xtrack tidal data and we used them for a validation of the tide reanalysis. The results are good and provide a validation in the open sea, not only near the coasts. See the new sections 2.2.2 and 3.2.1.

- *In the perturbation runs, why the drag coefficient Cd in the quadratic formulation is not perturbed? Dissipation of energy with the scales smaller than tides through the bottom friction could also be an important process that determines the characteristics of tidal currents, and in this sense, although the authors stated that the current research is focusing on SLA variations, in the current configuration, accuracy in flows will also be an important aspect.*

This part has been corrected, the perturbation of Cd and of the loading tide parameter (that we forgot to expose) are now described in section 2.3.1, row 215.

*Did the authors analyze whether the current design could also improve flows or not?*

We noted a change in water transports compared to the simulation without DA but we did not compare them to any measures. However, if the cross-correlation between levels and currents is correct (the size of the ensemble - 81 members - should be sufficient), then currents should improve as well. On a smaller scale, we had seen improvements in the current by assimilating sea-level data in the inlets of the Lagoon of Venice (Ferrarin et al., 2021).

- *In my opinion, it is still important to rely on DA to improve the parameterization in the simulation, since it is not that feasible for operational users to generate a large number of perturbation runs to have that short-term forecast improved.*

We added a sentence on parameter estimation as possible improvement in the conclusions (rows 542-547). We also added a part in the discussion with the computational speed of the current system in a daily simulation (about 30 minutes), with a detailed explanation (rows 494-500).

- *It is really hard to intensify the meshes in Figure 1. Could you zoom in to some critically locations to show the spatial variability of resolution?*

We changed the figure, adding a zoom in the northern Adriatic.

*Reviewer 2*
*The manuscript presents the predictive capability of a 2D barotropic model of the Mediterranean Sea sea level with and without the assimilation of the observations obtained from coastal tide gauges stations. The hydrodynamical model setup and ensemble Kalman filter based data assimilation system is described along with the perturbation schemes applied for ensemble generation. The results are presented for the total sea level as well as different contributions from the astronomical tides, surge and seiche for the hindcast/analysis and forecast periods for the December 2019 seiche occurrence following the November 2019 extreme event in the Adriatic Sea.*

*The manuscript requires a substantial revision before publication. Below are major comments and minor suggestions.*

We thank the Reviewer for the dedicated time and detailed review, which helps to increase the quality of this work. Below we provide the answer to the individual points.

*Major comments*

*To start with, for the readability of the manuscript, I suggest including a table of experiments to make it easier to follow, especially the results section. A flow chart for the production cycle would also help since it is difficult to understand where the hindcast/analysis ends and where the forecast starts. This may also help for future works since this system is proposed as a candidate for operational forecasting.*

As suggested, we added a Table with all the simulations and their characteristics (Tab. 1). We also set identification labels to call in the paper. Then, we added also a flow diagram with the forecast cycle (reanalysis cycle is simple), similar to the ones used in the CMEMS manuals of the models (Fig. 2).

*Moreover, the terminology used can be improved. There are terms used interchangeably such as analysis, reanalysis, hindcast simulation with data assimilation. I suggest homogenising them for an easier read and paying attention throughout the text to use the terminology that is already established, such as using analysis ensemble mean instead of average analysis state.*

We checked all the paper, to use a coherent terminology. Now hindcast simulations are the two-month simulations without DA, reanalysis are the two-month simulations with DA, forecast simulations are the forecast simulations, starting or from the background state (no DA) or from the analysis state (DA). See the introduction (rows 53-58) and the section 2.4 (rows 240-260). We also used the definition *analysis ensemble mean* as suggested.

*Secondly, I understand that the manuscript targets seiche in December 2019 however, it would be nice to see the evolution of the error in the sea level over a longer period given that the current version of the model is quite cheap as stated by the authors. I expected at least to see some analysis and the skill of the model in the November 2019 high tide event in the northern Adriatic Sea which resulted in the flooding of the city of Venice.*

We are planning to run reanalysis simulations for several years and we wrote this in the conclusions. Moreover, as suggested, we added the description and the reproduction of the November 2019's storm surge event (which was one of the most extremes). This event is described in section 3.3.1, with some plots. The results show that the wind forcing was bad one day before, but it was good the same day of the event. As

expected, DA does not improve the forecast, since that storm surge event was mainly due to the wind and pressure forcing. However, the storm surge of the day after (wich was still extreme) was forcasted much better with DA, as there was a component of seiche (Fig. 10).

*On the other hand, SHYFEM is shown to be a skillful model in various previous studies. It is hard to understand why a simplified version is used in a development that is a candidate for an operational forecasting system. I think that in the cases where the errors and bias are large there is missing the steric steric part from the thermohaline contribution to sea level variability. This should be clarified and justified.*

We specified better in the introduction that the focus of this paper is on the barotropic components (tides, surges and seiches), as specified in the title (rows 58-60).

Although the use of a barotropic model is a consequence of this, we added also a part, in the model description, with several works on tides, surges and seiches where we used a similar configuration. We also cited many works of other authors that use similar configurations with different models (rows 112-124).

*Finally, it is not easy to completely grasp the improvements brought by the data assimilation of observations from tide gauges since they are limited in space coverage. Satellite observations could be used at least for validation to see the impact, if not assimilated. The results should be supported by maps of, for example, mean dynamic topography, increments. I think there may be other resources for the coastal sea level data for assimilation such as Copernicus Marine, SeaDataNet or EMODNet to better cover the eastern basin.*

Based on this comment and on a comment of the first Reviewer, we looked for altimeter data to use for the validation. We downloaded the AVISO X-track data of amplitudes and phases of the tidal constants along the satellite tracks. From these constants, we computed the tidal sea level and we compared it with the model one. The average CRMSE decreases from 11.6cm to 4.3cm. See the new sections 2.2.2 and 3.2.1 and the new figure 5.

*Minor suggestions*

*Title: Mediterranean -> Mediterranean Sea*

Ok.

*L27 "easily predictable" -> please refer to the sources of uncertainty in the estimates of tides e.g. bathymetry*

Ok (rows 43-46).

*L92 Please be more precise about the mesh resolution and give a measure of change from the open ocean to the coastal seas. Danilov (2022) may help. https://agupubs.onlinelibrary.wiley.com/doi/full/10.1029/2022MS003177*

Ok (row 109-112 and Fig.1).

*L101 as done with the atmospheric forcing product, please cite the Copernicus Marine multi-year product explicitly in the references, not only with DOI. It should be clarified*

*why the authors used the multi-year product for the lateral open boundary conditions in the Atlantic Ocean while a NRT analysis/forecast product as in the atmospheric forcing is available in Copernicus Marine catalogs with tides for the experiment period. This is also one of the parameters that defines the type of experiment performed: an analysis, a reanalysis etc…*

The DOI in the paper was wrong, actually we used the analysis/forecast product that the Reviewer cites. Now we cite the right product, with the paper and the right DOI (rows 135-142).

*L102 please explain how you de-tide the sea level.*

As we wrote in the previous point, we put the wrong product. Actually, the de-tided sea level is provided in the model's variables (rows 138-139).

*L124 missing citation in the parentheses. Please add it.*

Done.

*L128 Please add the mean sea level map and compare with the MDT products such as MDT-CMEMS_2020_MED  in Copernicus Marine Catalog*

This sentence was badly written. Actually, the MDT products are different from the MSL of the model. Now we removed the word "MDT" referred to the model. Now the sentence should be clearer, we used a similar approach as Byrne 2021 (rows 159-163).

*L145 please justify 2 cm of observational error, is it only the instrumental error considered? How do the increments with such a small observational error look like? A map of increments may help to see whether there is an overfitting.*

The stations' sensors have an accuracy much lower than 1cm (radar sensors, see e.g.: https://www.mareografico.it/?session=0S1476768288B907168WO8287&syslng=ita&sysmen=-1&sysind=-1&syssub=-1&sysfnt=0&code=SENS&idse=C). Some have pressure sensors with an error of about 1cm.

However, in order to obtain the best results we tested several values (1,2,3 cm). We did not write this before, now we explain better this and other DA settings (rows 272-290).

*L153 grid -> node*

Ok.

*L153 "A_a^* is that of the analysis states not corrected". What do you mean? The definition of analysis implies a corrected background. Do you mean background?*

This part was badly explained. Now it is written better (row 189-199).

*L156 "levels" -> of what?*

The sea level in the model equations $\zeta$ in equations 1 (row 201).

*L162 "average analysis state" -> analysis ensemble mean*

Ok, we changed it in the whole paper.

*L169 Please justify 400 km. For example, Sakov et al. 2012 chose 250 km in a north Atlantic - Arctic Ocean system using the same methodology.*

As in Sakov et al. (2012), our justification is mainly empirical (we tried also different values). We saw that 400km gives good results. Anyway, 400km SLP perturbations produce sub-synoptic systems that are of the same length scale as the typical ones in the Mediterranean Sea. We added a better explanation (and we corrected the pressure std value which was wrong) (rows 222-225).

*L192 This is the definition of analysis ensemble mean. Please use it.*

Ok, we checked all the paper to use the right definitions.

*L 195 Not clear what the discussion here is.*

This concept is now explained better and moved in the introduction (rows 60-64).

*L 201 Why brevity? Why not robustness?*

This sentence was not clear, now the text has changed.

*L202-206 a production cycle flow chart may help.*

We made it as suggested (Fig.2).

*L222 What are the parameters of DA? Inflation and localization?*

This is now explained better. In the section 2.3 the general description and in 3.1 the final values of the DA parameters.

*L223 Local analysis is only one way of localization.*

The text was wrong it is removed.

*L234 Looks like too big error (9.3 cm) even for a free model and with a 2 cm of observation error reduces to only 3.6 cm. Is it because the barotropic model is missing the steric contribution? Please compare with altimeter products.*

Actually, as the Reviewer noted, such error could be partly due to the fact that the model without DA is missing the steric contribution and partly to the fact that November-December 2019 was a period in which also the barotropic part of the mean sea level was high. The DA corrects both.

About the altimeter products, now we used them for the tide validation (sections 2.2.2 and 3.2.1).

*L347 "is not present in our observations"? Do you mean in the period of observations used?*

Yes, we wrote better this sentence (row 439). We also added an explanation for the peak at 5.2h visible in some Adriatic stations (Fig.?), and we found a recent paper (Sepic, 2022) citing it (rows 441-443, 454).

*L352 There are other sources of error in DA besides model and representativeness error. Please correct.*

This sentence is off topic here and we removed it.

---

## Author Response (AR2)

Reviewer 2:

**The authors improved the manuscript substantially. It now is more complete and consistent in itself. I thank them for carefully addressing my comments in the first round. Though I'm still willing to see the maps of ssh increments at least for one assimilation cycle, in the lack of localization and assimilating only coastal observations. I couldn't find them in the revised version even though I asked for it. I suggest a major revision only to see the figure before acceptance. Thanks.**

*We thank the Reviewer for the time dedicated to improving our paper. Actually, we forgot the question about the analysis increments. Now, we made a new figure (n. 13), which shows the spatial distribution of the increments of the ensemble mean of the analysis with respect to the background. The map, as suggested, is the average of a daily assimilation cycle (24-time steps). The result is interesting and is discussed in the lines 391-398.*
*We also went through the paper, improving English and fluency and eliminating some redundancies.*